# INVESTIGATING ONLINE RL IN WORLD MODELS

## ABSTRACT

Significant advances in online reinforcement learning (RL) remain limited by the need for extensive environment interaction or accurate simulators. World models trained on large-scale uncurated offline data could provide a training paradigm for generalist AI agents which alleviates the need for task specific simulation environments. Unfortunately, current offline RL methods rely on truncated rollouts that can lead to value overestimation and limit out-of-sample exploration. Additioanlly, common offline RL datasets have been shows to have a bias towards healthy behavior which does not help with the development of generalizable methods. We propose an algorithm and a data curation method that addresses both of these concerns by demonstrating that effective full-length rollout training is possible *without hand-crafted penalties* by treating each member of the world model ensemble as a level in the Unsupervised Environment Design (UED) framework. Our method achieves competitive performance even with less transitions than the same online algorithms are traditionally trained on. We find that training a recurrent policy on an ensemble of world models is sufficient to ensure transfer to the original environment and match online PPO performance on standard offline-RL benchmarks while maintaining robust performance on our dataset, where conventional offline RL methods underperform. [1]

## 1 INTRODUCTION

Exploiting large amounts of data has proven to be a crucial component of recent advancements in machine learning. Generative models across multiple modalities—such as large language models (e.g., (OpenAI et al., 2024; Touvron et al., 2023)), text-to-image models (e.g., (Imagen-Team-Google et al., 2024; Betker et al., 2023)), and text-to-video models (e.g., (Brooks et al., 2024))—demonstrate that scale and coverage often outweigh the benefits of curation or the injection of favorable biases.

Reinforcement Learning (RL) (Sutton & Barto, 2018) has shown great promise in solving complex problems whenever *fast and accurate* simulation environments are available, such as in computer games (Silver et al., 2016a). Unfortunately, reliance on simulators has severely limited the applicability of RL to real-world problem settings. World models (Ha & Schmidhuber, 2018) offer a solution by learning approximate dynamics models from state transitions data and reducing reliance on task-specific hand-coded simulators. While increasing the dataset size can improve the fidelity of learned world models, they are rarely perfect recreations of the underlying environment. Ha & Schmidhuber (2018) demonstrate how RL agents frequently learn to exploit discontinuities and edge cases in *learned* dynamics to receive large spikes in simulated reward while learning unhelpful behaviors for the true environment.

This is problem is also addressed in *offline RL*, where the goal is to produce high-performing policies based only on a *static offline dataset* without any training signal from the real environment. Offline RL methods employ several algorithmic tricks to regularize learning towards the offline data distribution and enforce conservatism (Kumar et al., 2020). These include severely *truncating rollouts* to only a handful of consecutive steps inside a dynamic model, and *uncertainty penalties* that discourage the agent from stepping into parts of the state space of high uncertainty as done in MOPO (Yu et al., 2020) and MOREL (Kidambi et al., 2020). Recent work by Sims et al. (2024) demonstrates that the short truncated rollouts prevent compounding errors and outperform model-free methods

---

[1]Anonymous repo: `https://anonymous.4open.science/r/OnlineRLinWorldModels`

at the cost of pathological overestimation for the states at the edge of truncation. The misconception regarding the effectiveness of truncated rollouts has persisted partially due to well-established benchmarks like D4RL (Fu et al., 2020) are fairly saturated (Sun, 2023) and have recently been shown to be biased towards healthy states and positive, near-optimal performance (Li et al., 2024).

While full-length rollouts can avoid the truncation pathologies, they are more susceptible to compounding error and world model exploitation that handicaps transfer to the real environment. For a solution, we turn to Unsupervised Environment Design (UED) (Dennis et al., 2020; Jiang et al., 2021b;a; Parker-Holder et al., 2022), a class of online RL methods that can address the need for zero-shot adaptations by training agents to be robust across varying train and test distributions. These methods seek to minimize maximum regret over a space of levels (Dennis et al., 2020). We break the traditionally constrained setting of UED and use it to select over a large number of world models trained on the same dataset with each models serving as a given *level* in UED.

Pathological algorithms and positively biased datasets impede training generalist RL agents by not making use of large amounts of data *and* recent advances in online RL. In this work, our contributions consist in: 1) investigating training through full-length offline rollouts to address model-based offline RL challenges, 2) produce a dataset that does not exhibit the biases in previous benchmarks, and 3) introduce the **P**olicy **O**ptimization with **W**orld **E**nsemble **R**ollouts (**POWER**) algorithm that utilizes several UED methods to select which world model the agent will interact with at every step. We show that our algorithm outperforms standard offline RL methods on our dataset while achieving comparable results to online PPO when trained offline on the D4RL dataset. Additionally, we demonstrate that our method produces diverse world models even when trained on the same data.

## 2 PRELIMINARIES

### 2.1 CONTEXTUAL MARKOV DECISION PROCESS

We define a infinite-horizon, discounted contextual Markov decision process (CMDP) (Hallak et al., 2015) by introducing a context variable $\theta \in \Theta \subseteq \mathbb{R}^d$:

$$\mathcal{M}(\theta) \coloneqq \langle \mathcal{S}, \mathcal{A}, P_0, P_S(s, a, \theta), P_R(s, a, \theta), \gamma \rangle, \tag{1}$$

where each $\theta$ indexes a specific MDP by parametrising a transition distribution $P_S(s, a, \theta) : \mathcal{S} \times \mathcal{A} \times \Theta \to \mathcal{P}(\mathcal{S})$ and reward distribution $P_R(s, a, \theta) : \mathcal{S} \times \mathcal{A} \times \Theta \to \mathcal{P}(\mathbb{R})$. We denote the corresponding joint conditional state-reward transition distribution as $P_{R,S}(s, a, \theta)$. Context variable $\theta$ can also be referred to as a *level*, terms that are used interchangeably in this paper.

At timestep $t$, an agent follows a policy $\pi : \mathcal{S} \times \Theta \to \mathcal{P}(\mathcal{A})$, taking actions $a_t \sim \pi(s_t, \theta)$. We denote the set of all context-conditioned policies as $\Pi_\Theta \coloneqq \{\pi : \mathcal{S} \times \Theta \to \mathcal{P}(\mathcal{A})\}$. The agent is assigned an initial state $s_0 \sim P_0$. As the agent interacts with the environment, it observes a history of data $h_t \coloneqq \{s_0, a_0, r_0, s_1, a_1, r_1, \ldots a_{t-1}, r_{t-1}, s_t\} \in \mathcal{H}_t$ where $\mathcal{H}_t$ is the corresponding state-action-reward product space. We denote the context-conditioned distribution over history $h_t$ as: $P_t^\pi(\theta)$ with density $p_t^\pi(h_t|\theta) = p_0(s_0) \prod_{i=0}^{t} \pi(a_i|s_i, \theta) p(r_i, s_{i+1}|s_i, a_i, \theta)$.

In the infinite-horizon, discounted setting, the goal of an agent in MDP $\mathcal{M}(\theta)$ is to find a policy that optimises the objective:

$$J^\pi(\theta) = \mathbb{E}_{\tau_\infty \sim P_\infty^\pi(\theta)} \left[ \sum_{t=0}^{\infty} \gamma^t r_t \right]. \tag{2}$$

We denote an optimal policy as $\pi^\star(\cdot, \theta) \in \Pi_\Theta^\star(\theta) \coloneqq \arg\max_{\pi \in \Pi_\Theta} J^\pi(\theta)$, where $\Pi_\Theta^\star(\theta)$ is the set of all optimal MDP-conditioned policies that are optimal for $\mathcal{M}(\theta)$.

### 2.2 UNSUPERVISED ENVIRONMENT DESIGN

Unsupervised environment design (UED) is a class of autocurriculum methods for RL, where an adversary proposes tasks for an agent to train on. Commonly (Dennis et al., 2020), environments are modelled as a CMDP $\mathcal{M}(\theta)$ (see Equation (1)) known as underspecified Markov decision process where each context $\theta \in \Theta$ is known as a level.

The recent approach of Minimax Regret (MMR) UED has emerged as a promising way to train robust agents (Dennis et al., 2020; Jiang et al., 2021b;a; Parker-Holder et al., 2022). Here, the

adversary chooses levels that maximise the agent's *regret*, defined as:

$$\text{Regret}_\theta(\pi) := J^{\pi^\star}(\theta) - J^\pi(\theta). \tag{3}$$

Dennis et al. (2020) posed the UED setting as a two-player, zero-sum game between the adversary and the policy. They show that if the adversary aims to maximize regret and is in Nash equilibrium with the policy, the following holds:

$$\pi_{\text{MinMax}} \in \arg\min_{\pi \in \Pi_{\mathcal{H}}}\{\max_{\theta \in \Theta}\{\text{Regret}_\theta(\pi)\}\}. \tag{4}$$

Minimizing the worst-case regret confers a degree of robustness to the policy as its regret in any level $\theta \in \Theta$ must be below this bound. See Appendix A.1 for a more detailed discussion.

### 2.2.1 PRIORITIZED LEVEL REPLAY

Prioritized Level Replay (Jiang et al., 2021b) is an empirically successful curriculum method that relies on curating high-scoring levels. In practice, PLR maintains a buffer of previous high-scoring levels, and either samples from this buffer, or samples new levels. The agent is rolled out on these new levels, and they are scored depending on its performance. High-scoring levels are added to the buffer, and the agent trains on the collected experience.

The original PLR scores each level $\theta_i$ using a time-averaged $L_1$ value loss of each agent's last trajectory on the level (Jiang et al., 2021b). In order to achieve *minimax robustness*, a scoring function should account for regret as described in Section 2.2. Jiang et al. (2021a) propose different scoring functions that more closely approximate the regret. Ultimately, the choice of a scoring function is a design choice depending on the nature of the environment. We further elaborate on the scoring function choices in section 3.

### 2.3 WORLD MODELS

As defined by Ha & Schmidhuber (2018), world models are representations of the dynamics of an environment. From an agent's perspective, a trained world model can be interacted with in the same way as the true environment. In this work, we implement the world model as a one-step dynamic model. World models are generally represented using a neural network that jointly parametrizes the transition distribution $P_{\mathcal{S}}$ and rewards distribution $P_{\mathcal{R}}$ from Equation (1). Therefore, we train $\mathcal{F}_\theta$ as $\mathcal{F}_\theta(\hat{s}_t, a_t) \rightarrow \hat{s}_{t+1}, \hat{r}_{t+1}$ by predicting *both* the state transition and the reward of the agent.

## 3 TRAINING WITH WORLD MODEL ENSEMBLE ROLLOUTS

We introduce **P**olicy **O**ptimization with **W**orld model **E**nsemble **R**ollouts (**POWER**), to leverage large datasets and benefit from effective methods used in traditionally online settings. As shown in Figure 1, we start by training a collection of world models consistent with the provided data. We then treat these models as *levels* and select them based on different sampling methods to train a transferable policy as outlined in Algorithm 1. Our implementations allows for the agent to see different world models within the same trajectory shown in Fig. 1(left) or *only one* per episode which is then used to score the *model's* likelihood of being sampled again in the course of training as shown in Fig. 1(right) and elaborated in 2.2.1.

### 3.1 TRAINING MULTIPLE WORLD MODELS

In this work, we assume access to a *non-sequential* offline dataset $\mathcal{D}$ of $N$ state-action-state-reward transition observations: $\mathcal{D} = \{(s_i, a_i, s'_i, r_i)\}_{i=0}^{N-1}$, all collected from a single MDP $\theta^\star$. We address this tractability issue by learning a highly informative posterior distribution using offline data, which concentrates around a small region of the parameter space $\Theta$ containing the true dynamics $\theta^\star$. By doing so, we effectively reduce the hypothesis space to a manageable subset of $\Theta$, enabling the tractable evaluation of the RL objective.

Practically, we implement this by training multiple distinct world models each initialized differently and trained on different permutations of the data. The inherent variability introduced by stochastic gradient descent during the training process causes each world model to exhibit slightly different

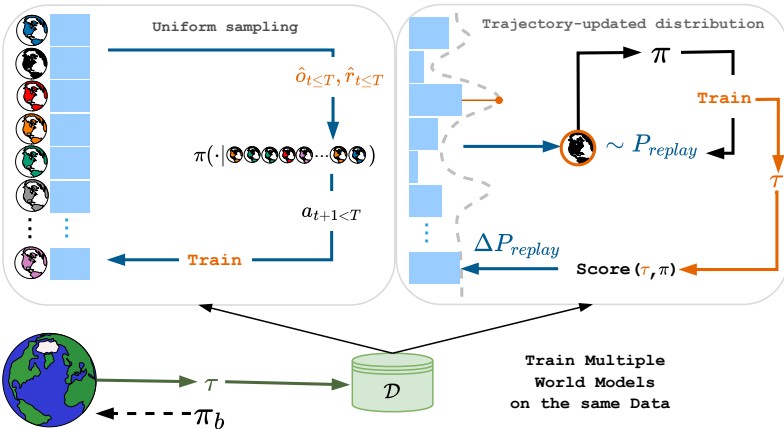

Figure 1: An overview of the two groups of sampling methods that can be selected. Our algorithm can allow for either sampling a new world model each step as illustrated by the `Uniform Sampling` block in the left or selecting *only* after a full trajectory finishes as done in UED methods illustrated by the `Trajectory-updated` block to the right.

dynamics (Amari, 1993). However, an agent trained in any one of these world models is not guaranteed to transfer well to the real environment, and it is this problem we tackle by using the ensemble of world models.

## 3.2 WORLD MODELS AS LEVELS

If we treat each world model $\theta$ as a *level*, we can apply standard minimax regret algorithms to our setting. More formally, we consider the two-player game between an adversary $G$ and student policy $\pi$, such that the adversary generates a level (i.e., a world model) $\theta \in \Theta$ that maximizes the agent's regret, and the agent trains as normal on the provided levels. Note, we define $\Theta \doteq \{\theta : L_2(\theta, \mathcal{D}) < \epsilon\}$ to be the set of all world models that have loss over the dataset $\mathcal{D}$ of less than some threshold $\epsilon$. At Nash equilibrium of this game, Dennis et al. (2020) showed that the policy satisfies Equation (4). In other words, the policy's maximum regret on any $\theta \in \Theta$ is bounded by $W \doteq \min_{\pi \in \Pi} \{\max_{\theta \in \Theta} \{\text{Regret}_\theta(\pi)\}\}$. Since we have assumed that $\theta^\star \in \Theta$, *this bound further applies to the true environment dynamics*. Moreover, since the adversary is constrained to only choose levels within $\Theta$, i.e., those that have loss less than a certain value, it cannot be overly adversarial and provide totally unrealistic dynamics to train the agent on.

**Algorithm 1** Policy Optimization with World Model Ensemble Rollouts (POWER) with PLR, DR or DR-Step

```
1: Inputs: Dataset D; model count M;
2: PLR flag; DR-Step flag
3: for i = 1 to M do
4:     Initialize θ_i ∼ N(0, σ²)          LeCun Normal
5:     Shuffle D to get D_i             Use different seeds
6:     Train θ_i on D_i to convergence      Use L2 loss
7: end for
8: π, h_t ← h_0                   Initialize recurrent policy
9: while π not converged do
10:    if PLR then
11:        i ∼ Sample Using PVL score S_i        use PLR
12:    else
13:        i ∼ U(1, M)                          use DR
14:    end if
15:    τ ← {}                       Initialize trajectory set
16:    s_0 ∼ P_0^{θ_i}             Initialize from learned P_0
17:    for t = 0 to T − 1 do               episode length T
18:        if DR-Step then
19:            i ∼ U(1, M)                   use DR-Step
20:        end if
21:        a_t ∼ π(·|h_t, s_t)               Sample action
22:        s_{t+1}, r_{t+1} ∼ F_{θ_i}(s_t, a_t)   Step in world model
23:        τ ← τ ∪ {(s_t, a_t, s_{t+1})}      Add transition
24:        h_{t+1} ← h_t ∪ {s_{t+1}}         Update hidden state
25:    end for
26:    Update π using τ                      PPO update
27:    Update PVL score S_i using Equation 5
28: end while
29: Output: π
```

In order to make this procedure practical, we use the high-performing PLR algorithm as illustrated in the right side in Figure 1, treating different world models $\theta$ as levels. Despite PLR not guaranteeing convergence to a Nash equilibrium, it generally results in improved zero-shot generalisation to out-of-distribution tasks. Since regret for a given world model is not always known, we use the standard regret approximations of Positive Value Loss for level $\theta_i$ where $\gamma$ and $\lambda$ are the MDP and GAE discount factors and $\delta_t$ is the TD-error at timestep $t$ as framed by (Sutton & Barto, 2018):

$$S_i = \frac{1}{T} \sum_{t=0}^{T} \max \left( \sum_{k=t}^{T} (\gamma\lambda)^{k-t} \delta_k, 0 \right). \tag{5}$$

## 4 EXPERIMENTAL SETUP

### 4.1 DATASET CURATION

Our dataset curation strategy is guided by the concept of *state coverage*. Using a single behavior policy $\pi_b$ often results in exploring a limited subset of the state space. To address this limitation, we employ multiple behavior policies to gather diverse data. Specifically, we train an agent in the real environment using Proximal Policy Optimization (PPO) (Schulman et al., 2017) and periodically create checkpoints throughout training to convergence. These checkpoints serve as distinct behavior policies, ensuring that our dataset encompasses a wide range of behaviors—from those generated by randomly initialized policies to those that effectively solve the task. Fu et al. (2020) point out that different dataset distributions can encourage conservative approaches or be more amenable to imitation learning and behavior cloning. Our dataset curation is agnostic to these tendencies.

We note that our dataset is shuffled in the level of state transitions and *does not* require sequences to train the world models. The frequency of checkpointing and the number of trajectories collected at each checkpoint are determined to match D4RL's orders of magnitude of no more than $10^6$ transitions. We stop collecting after one or two convergence checkpoint in order to not bias our dataset. Figure 2 demonstrates the schedule for collecting behavior policy trajectories in the Hopper environment. A.2 contains the sizes for each environment.

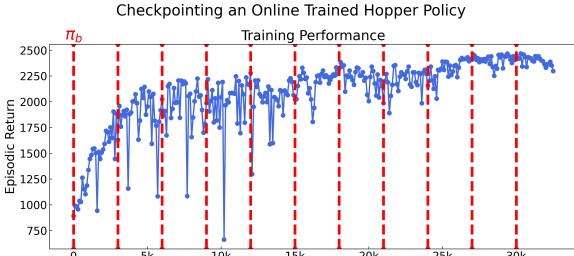

Figure 2: Collection of dataset $\mathcal{D}$ using different $\pi_b$ checkpoints marked by the vertical lines.

### 4.2 WORLD MODEL TRAINING

The world models are trained on the same data as described in line 4 to 6 of Algorithm 1. These models show different final test losses and therefore slightly different dynamics through the trajectory. The world models in our experiments are implemented as fully connected networks with a concatenated *input* of actions and observations and an *output* of the concatenated next observations and reward. With our method being agnostic to the architecture used for the ensemble, we also implement a *visual world model*. The fully-connected forward dynamics are kept the same with a standard convolution layer added to encode the visual observation in the beginning and a downstream decoder to reconstruct the output to the shape of the observations used by the agent. The models are trained in parallel using vmap – a vectorizing map possible through our JAX-based implementation (Heek et al., 2024). We advise caution with the number of visual world models trained in parallel given the dimensions of the pixel-based input. We design our implementation to require only a single GPU.

Refer to A.3 for an overview of the computational efficiency that allows the training of the multiple world models in parallel, A.4 for the test performance and A.5 for the hyperparameters.

### 4.3 Training the Reinforcement Learning Agent

We use a recurrent actor-critic network based on PureJaxRL (Lu et al., 2022) and the convolution actor-critic from (Becktepe et al., 2024) for the visual agent. The agent's actions depend on the current observation and interaction history, implemented as the recurrent state of the actor-critic network. We use the recurrent state to test the agent's ability to perform system identification across the world models it is trained on. This is also done to verify that the world models have distinct dynamics.

The configurations passed at the start of our algorithm 1 as boolean flags allow for the following set of world model selection methods to be tested:

**PLR**: Prioritized Level Replay as described in with an $L_1$ value loss score function as done in the original paper by Jiang et al. (2021b). **PLR_PVL** uses Positive Value Loss scoring in Equation 5. Used by setting *only* the PLR flag to True in Algorithm 1.

**DR**: Domain Randomization implemented by randomly selecting a new world model $\theta$ from a uniform distribution over the trained world models as done in line 13 of our algorithm. Used by setting *both* the PLR and DR-Step flags to False in Algorithm 1.

**DR-STEP**: Change $\theta_i$ for every step of the agent in a fixed length episode instead of only doing it at the start of a trajectory. Used by setting *only* the DR-Step flag to True in Algorithm 1.

**DR-PROB**: A simple change in line 19 of our algorithm to either perform **DR-STEP** or not change $\theta_i$ with probability $p$. The probability $p$ could also serve as a classic UED parameter where p is varied based on the episode's score. Such use is, however, outside the scope of this work.

**WM**: *A single world model* $\theta_i$ for the entire training, all flags set to False and the model is sampled *only* once when the policy is initialized.

To address policy overfitting to the world models' dynamics without querying the real environment during training, we hold out world models trained on transitions from the test set used for the world models training. We observe that when overfitting occurs, as indicated by the decoupling of training and evaluation rewards, the standard deviation of the policy across the holdout world models increases. This phenomenon serves as a reliable indicator for early stopping and helps prevent policy overfitting. We note that our method and hyperparameters do not rely on online tuning.

The PLR implementations are based on JaxUED (Coward et al., 2024). We use the RLiable library (Agarwal et al., 2021) to measure the performance. Every metric is plotted within a 95% confidence interval calculated over five seeds and 50 episodes on the respective environment. Our entire work is implemented in the JAX Ecosystem (DeepMind et al., 2020) for end-to-end GPU training.

### 4.4 Baselines

We baseline our methods by training on a randomly sampled single world model (WM) and against commonplace offline RL algorithms like CQL (Kumar et al., 2020) and SACn (An et al., 2021).

Our implementation is based on the CORL (Tarasov et al., 2022) and its JAX port (Nishimori, 2024). We verified the implementation's correctness and hyperparameters to reproduce the reported performance on Halfcheetah and Hopper D4RL datasets. We then performed a grid search over our own dataset to record the highest score obtained by the baselines. While our method only requires single-step transitions, we maintained fairness in comparison with CQL and SACn for the lower ratios by downsampling episodes uniformly rather than individual transitions, as both CQL and SACn were designed to operate on complete trajectories. The specific ranges can be found on Table 8 and Table 9.

## 5 Results

In this section we show the most notable results that elucidate important aspect of our approach. A complete compilation of the results can be found in the Appendix. We collect data from and evaluate on environments from the Gymnax (Lange, 2022) and Brax (Freeman et al., 2021) suites. All the

evaluations are performed on full trajectories across five random seeds on the corresponding real environments.

## 5.1 PREVENTING EXPLOITATION

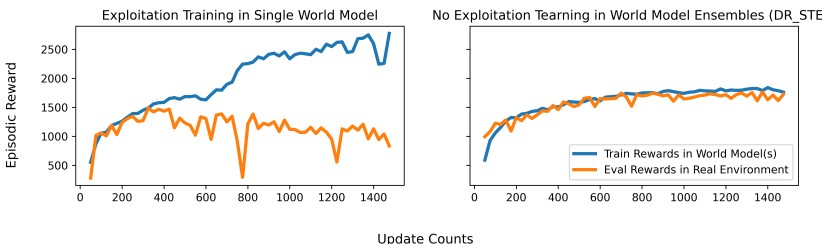

Figure 3: Preventing reward hijacking of the learned model by using the ensemble training method

Training in world model ensembles prevents the agents from overfitting to the training distribution and hacking the rewards. Figure 3 shows the results on a world models trained with $2 \cdot 10^4$ transitions, only 20 episodes worth of transitions.

## 5.2 CLASSIC CONTROL

The suite of methods using world model ensembles outperforms naive world model training with only a couple of episodes worth of transitions from dataset $\mathcal{D}$. We illustrate the evaluation on the Cartpole environment in Figure 4 to showcase the effectiveness of world model ensembles to reach the highest episodic return possible in less than half the transition counts compared to using a single world model. Training on multiple world models beats the single world models baseline in a simple environment. Figure 5 shows our methods consistently outperform training on a single world model for sparser data and even achieve returns higher than the behavior policy that was learned online. Figure 8 shows the comparison with model-free offline methods for pendulum.

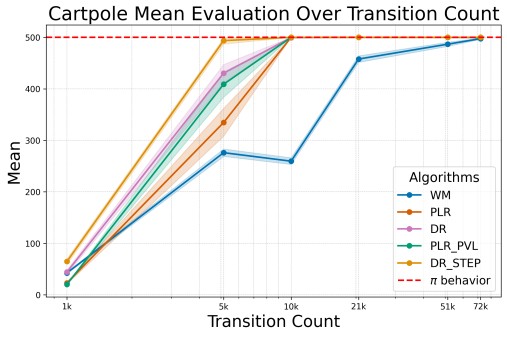

Figure 4: Mean of the evaluations on Cartpole

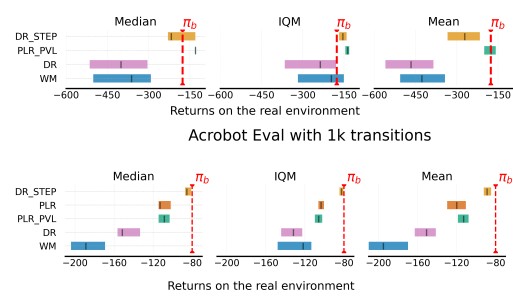

Figure 5: Interquartile Mean (IQM), Mean, and Median of the world model ensemble trained policy evaluated on the real environment

## 5.3 RESULTS BRAX WITH OUR DATASETS

We test our algorithms and its variations on Hopper (Figure 6) and Halfcheetah (Figure 7) from the Brax suite of environment. We notice that the methods that sample a new level uniformly at every step or with a probability $p$ outperform every method in sparser data regimes.

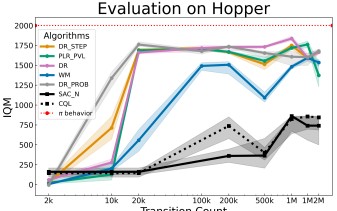 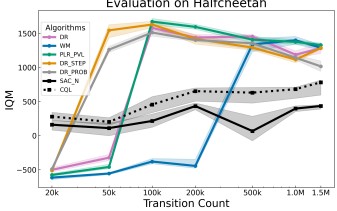 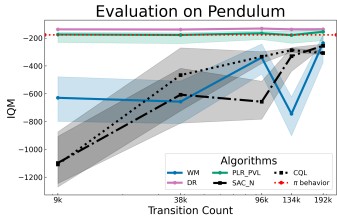

Figure 6: IQM for Hopper  Figure 7: IQM for Half Cheetah  Figure 8: IQM for Pendulum

## 5.4 RESULTS IN MUJOCO USING D4RL DATASETS

When applied to D4RL transitions, POWER and its variations achieve comparable performance to online PPO implementations (Figure 9) such as CleanRL and Stable Baselines (Huang et al., 2022). We chose PPO as our baseline since it is the same algorithm used within our world model ensemble using 1.

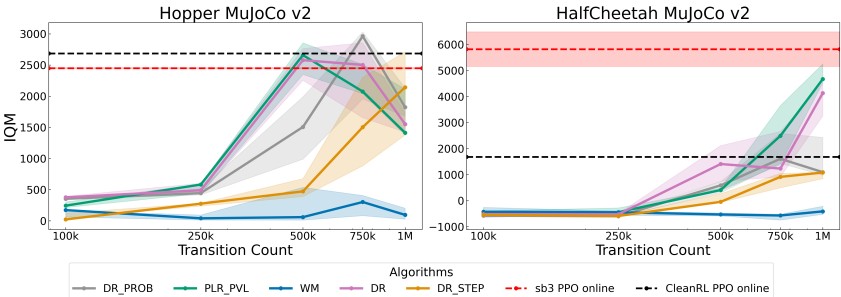

Figure 9: Results in MuJoCo using the D4RL dataset to train the world models, standard error over 5 seeds

## 5.5 ABLATING THE ENSEMBLE SIZE

We perform ablations across different variations of our method on the Hopper `full-replay-v2` dataset. The results demonstrate that while increasing the number of world models improves performance, we achieve strong results even with a relatively small ensemble size. This suggests that our approach effectively balances performance gains with computational efficiency, as significant benefits can be realized without requiring a large number of models.

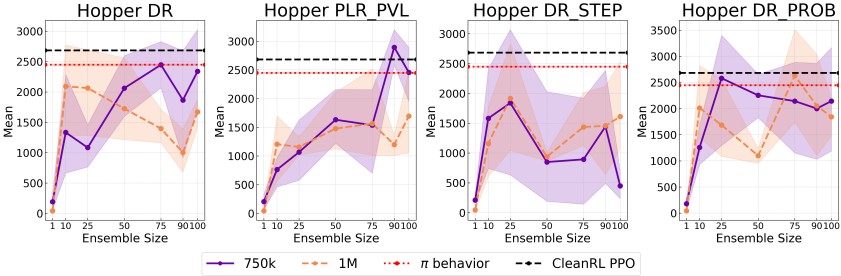

Figure 10: Ensemble size ablations for MuJoCo Hopper

Classical control ablations can be found in A.6.

## 5.6 RNN ANALYSIS

Our claim is that the world models have sufficiently distinct dynamics and can therefore serve as different contextual MDPs. If true, regret-based training should help the agent adapt to all these

dynamics. We demonstrate this by deploying our agent across multiple world models and on the real environment. We then train a classifier on the recurrent states of said agent to identify its environment and achieve an average of 62% accuracy on the **DR**, 60% on **PLR** and 45% on **PLR_PVL**; all above the 10% random prediction accuracy. More qualitative analysis in A.7 and classification results in A.8.

# 6 DISCUSSION

## 6.1 DATASET DISTRIBUTIONS

While our method achieves competitive results in world models trained on our dataset with wide state coverage, and our online PPO in world models matches the results of online PPO in the real respective environment, we do not reach the maximum D4RL scores other than with Hopper. We present the following investigation into why that is the case and why we think this points out to inherent biases in the field of offline RL that stand in the way of making use of data on the larger scale.

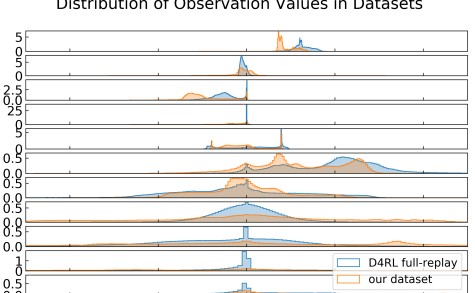

Figure 11: Observation Distribution in Hopper-full-replay datasets from D4RL and in ours

Figure 12: Action Distribution in Hopper-full-replay datasets and in ours

We reiterate that previous work Li et al. (2024) has shown that offline RL methods are susceptible to implicit biases in the data collection practice. Figure 11 offers a succinct qualitative analysis by showing that more than half of the Hopper dimensions from D4RL have narrower coverage and bias the agent towards healthy behavior; a helpful addition for Hopper as the unhealthy state flag can cause an early termination and vastly affect evaluation. This is even more significant when it comes to Walker2D where even online PPO underperforms Huang et al. (2022) compared to off-policy methods like SAC. A method that includes a Behavior Cloning term like TD3+BC (Fujimoto & Gu, 2021) is at a clear advantage since it is directly biased away from unhealthy states that would otherwise be explored more in the online environment (as our dataset distribution shows in Figures 11 through 12. The state of offline RL and its benchmarks has *positively reinforced* a direction of methods that does not account for the type increasingly available large scale datasets.

## 6.2 FUTURE WORK

Our work would benefit from a more principled and interpretable method of sampling the possible world models from Θ set – as defined in 3.2 – other than simply changing the shuffling and initialization seeds. A natural extension is that of level generation to have an expanding buffer of available levels during the adversarial training. Our method also offers a way to generate an RL training curricula by abstracting away hand-crafted heuristics and using data to generate different levels directly.

Finally, the results in physical engines like Brax should be extended to *real* physical platforms and address the engineering challenges posed by the *sim2real* gap, especially in sensitive settings where online training can be physically hazardous.

## 7    RELATED WORK

Reinforcement Learning has achieved impressive results, some of the most notable ones being Go (Silver et al., 2016b), Starcraft (Vinyals et al., 2019), Atari (Mnih et al., 2015) and more recent advances focusing on multi-task generalizations (Bruce et al., 2024; Hafner et al., 2023). Despite these impressive results, RL methods fail to generalize to settings even slightly different than the training environments (Cobbe et al., 2019; Mediratta et al., 2023), indicating that the generalization to real world settings remains an open challenge.

An RL agent can be more generalizable if exposed to a sufficiently diverse set of environments in training time. The Unsupervised Environment Design (UED) (Dennis et al., 2020; Jiang et al., 2021a) line of work achieves this by relaxing the definition of the environment to a combinatorially large set of possible configurations captured by a set of parameters, commonly referred to as *levels*. The choice of the parameter space is specifically tailored to the general task domain also known as the underspecified environments (e.g. a maze environment is parameterized by the placement of the walls, start and goal position whereas a one dimensional bipedal environment is parameterized by the roughness of the terrain). UED uses Minimax regret (Savage, 1951) to make the agent robust to the most challenging environment configurations without prior knowledge of which set of parameters it will act in. While these approaches are meant to exemplify deployment in challenging situations, they remain reliant on semantically informed choices of parameters to capture *levels* of difficulty.

World models (Ha & Schmidhuber, 2018) propose a different approach where the agent is equipped with a compact representation of the real environments trained using a dataset of transitions in said environment. More recent work shows that world models can serve as task-agnostic Continual Reinforcement Learning baselines (Kessler et al., 2023) or used in online RL to achieve human-level performance on Atari (Hafner et al., 2020). In principles, world modelling does not hinge on task-specific heuristics and only relies on increasing the robustness of the agent by tuning the uncertainty inside the world model. A recent combination of the world model and *Minimax Regret* approach by Rigter et al. (2023) trains a world model that can derive robust policies. This is done through an exploration policy seeking maximal model uncertainty, similar to the self-supervised world model methods by Sekar et al. (2020). These are ultimately online methods and require sufficient exploration of states that can be physically dangerous to the agent and disrupt operation altogether (Kumar et al., 2020; 2021).

Offline RL work has provided a useful signal on the importance of using offline datasets (Kumar et al., 2020; 2021), the common challenges that arise form the distribution shift between the behavior and learned policy (Levine et al., 2020) and model error (Saleh et al., 2022) alongside the most common workarounds like truncated rollouts (Jackson et al., 2024). Model-based offline (Rigter et al., 2022) and online (Chua et al., 2018) RL methods have served as useful blueprints to manage uncertainty through *multiple* dynamic models. Sims et al. (2024) demonstrated that short rollouts (1-5 steps) can cause pathological value estimation and algorithm collapse, emphasizing the importance of full-length trajectories. Additionally, Li et al. (2024) identified inherent biases in D4RL benchmarks, suggesting that methods relying on hand-crafted behavior cloning and conservative conditions may lack generalizability. These have been very useful signals in developing an approach not reliant on traditional offline RL tricks.

Finally, the work of Li & Liang (2018) and the foundational work of Amari (1993) have paved the intuition that shuffling the data and most importantly, changing the initializations, would be effective in training sufficiently distinct models *on the same dataset*.

## 8    CONCLUSION

In this work we present a novel way to guarantee transfer robustness to the real environment over world models fitted on offline data. To the best of our knowledge, this is the first work that performs adversarial training under this specific fully parametric constraint. The introduced algorithm and world mode selection enables the use of online-RL innovations in more general setting i.e. from grid world and simple environments to any problem there are transitions for. Our method naturally lends itself to other architectures and hopefully will help blaze the trails towards meaningful deployment of state-of-the-art RL algorithms into the *real* world based on training inside large scale generative models.

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

# A    APPENDIX

## A.1    UED DISCUSSION

In this section we revisit the main principles of UED and their connection to Bayesian RL. Our derivation reveals that minimax UED is equivalent to learning a Bayes-optimal policy under a least favourable prior. As Bayesian RL is a more general framework that allows for optimality under different priors, we now discuss the relative advantages and disadvantages of choosing a least favourable prior. The benefits of choosing a least favourable prior include:

**I. Policies are robust to changes in prior**    A key advantage of the least favourable prior is that policies can be robust to changes in belief. When the minimax theorem (Neumann, 1928) holds, a Nash equilibrium to the two-player game exists with solution $(\pi_{\text{MinMax}}, \Theta_{\max}^{\pi_{\text{MinMax}}})$ and it follows (Buening et al., 2023):

$$\min_{\pi \in \Pi_{\mathcal{H}}} \max_{\theta \in \Theta} \left[\text{Regret}_\theta(\pi)\right] = \min_{\pi \in \Pi_{\mathcal{H}}} \max_{P \in \mathcal{P}} \mathbb{E}_{\theta \sim P}\left[\text{Regret}_\theta(\pi)\right] = \max_{P \in \mathcal{P}} \min_{\pi \in \Pi_{\mathcal{H}}} \mathbb{E}_{\theta \sim P}\left[\text{Regret}_\theta(\pi)\right], \quad (6)$$

which implies that the minimax policy is robust to any change in the prior.

**II. Protection against worst case MDPs**    The set $\Theta_{\max}^{\pi_{\text{MinMax}}}$ indexes MDPs where policies have the worst possible regret. This ensures that the agent following $\pi_{\text{MinMax}}$ at test time is protected against situations where the return has the potential to be very low. From a safety perspective, this can protect an agent from behaving in a way that is dangerous towards itself or others in an environment; in particular, if an agent is at a Nash equilibrium, the regret across all MDPs is bounded by $\min_{\pi \in \Pi_{\mathcal{H}}} \max_{\theta \in \Theta} \left[\text{Regret}_\theta(\pi)\right]$.

There are also several drawbacks to choosing a least favourable prior. Many of these stem from the restriction of the prior to $\Theta_{\max}^{\pi_{\text{MinMax}}}$, and include:

**I. Inability to exploit prior knowledge**    The least favourable prior excludes the ability to integrate pre-existing beliefs into the Bayes-optimal policy. If prior knowledge about the set of environments is available, for example from and offline dataset or known skills that are common across all environments, this information cannot be exploited by a least favourable prior. This is most pertinent if the true distribution over context variables is known a priori, as using this as the prior results in the greatest regret reduction according to the frequency in which MDPs are encountered in practice.

**II. Inability to learn optimal policies**    For proper priors with support over $\Theta$, provided $\theta^\star \in \Theta$, a key property of Bayes-optimal policies is that they tend towards the optimal policy $\pi(s_t, \theta^\star)$ in the limit of $t \to \infty$. If the index $\theta^\star$ of true MDP allocated to the agent at test time lies outside of the set of worst regret parameters, that is $\theta^\star \notin \Theta_{\max}^{\pi_{\text{MinMax}}}$, then the posterior under the least favourable prior cannot collapse to place its support on $\theta^\star$ and the corresponding policy will never be optimal for $\mathcal{M}(\theta^\star)$. As $\Theta_{\max}^{\pi_{\text{MinMax}}}$ is typically a very small subset of $\Theta$ and the whole of $\Theta_{\max}^{\pi_{\text{MinMax}}}$ is never learned in practice, we expect this situation to be frequently encountered. This point has been observed empirically as the inability to generalise to out of distribution tasks (Jiang et al., 2021a).

**III. Issues with learning Nash equilibria**    The conditions needed to prove the existence of the minimax solution - a finite state-action space, a finite horizon, known reward, a finite set of MPDs (see Buening et al. (2023) for details) - rarely hold in a CMDP in practice. Whilst it is currently unknown whether the minimax theorem can be generalised to more realistic CMDPs, empirical evidence suggests this is not the case (Buening et al., 2023). MDPs where the Nash equilibrium does not exist present a convergence issue when learning a minimax policy. Moreover, even if the Nash equilibrium exists, algorithms rarely learn the entirety of $\Theta_{\max}^{\pi_{\text{MinMax}}}$ required for the minimax policy (Beukman et al., 2024). In particular, if the algorithm collapses to a prior with support over single context variable, we cannot expect the minimax policy to learn anything useful at test time.

**IV. Inherent pessimism**    A least favourable prior encodes the most pessimistic belief possible - that an agent will always be faced with a set of MDPs that have the potential for the highest regret. The agent does not consider any hypothesis outside of $\Theta_{\max}^{\pi_{\text{MinMax}}}$ when reasoning about its beliefs, despite the fact these MPDs may be more typical of the environments encountered at test time. This

prevents exploration of alternative hypotheses and is not a universally appropriate belief for every CMDP.

**V. Loss of admissibility**    A key benefit of Bayes-optimal policies is that, given a proper prior, they are guaranteed to be admissible - they cannot be Pareto improved upon in terms of expected return $J^\pi(\theta)$ across $\Theta$ (Wald, 1947; 1950). Least favourable priors are not guarenteed to be proper and there exist known counterexamples where inadmissible decisions are taken under a minimax policy.

**VI. Amplifying effects of model misspecification**    In most learning settings, it is not reasonable to assume that the practitioner can specify a CMPD that contains the exact space of MDPs that an agent could encounter. We must account for some degree of misspecification where there exist subsets of context variables $\Theta' \subset \Theta$ that do not correspond to a realisable model. By restricting the prior to have support over $\Theta_{\max}^{\pi_{\mathrm{MinMax}}}$, it may occur that the prior only has support over MDPs in $\Theta'$, hence the corresponding minimax policy will only account for MDPs that do not exist in practice.

Like any prior, we see that choice of using a least favourable prior is *subjective*, and its justification depends on weighing up the relative advantages and disadvantages by a practitioner on a case-by-case basis. Either way, the least favourable prior and minimax solution is by no means a universally appropriate method.

## A.2   DATASET SIZES

Here are transitions counts for each dataset. We use `full-replay` dataset for the D4RL experiments as those match our data curation strategy 4.1 the closest and have the widest state coverage.

Table 1: Transition Counts for each dataset

| Environment | Transition Count |
|---|---|
| Acrobot | $1.02 \cdot 10^5$ |
| Cartpole | $1.02 \cdot 10^5$ |
| Mountaincar | $1.03 \cdot 10^5$ |
| Pendulum | $1.92 \cdot 10^5$ |
| Hopper Brax | $2 \cdot 10^6$ |
| Halfcheetah Brax | $2 \cdot 10^6$ |
| Hopper D4RL | $1 \cdot 10^6$ |
| Halfcheetah D4RL | $1 \cdot 10^6$ |
| Walker2D D4RL | $1 \cdot 10^6$ |

## A.3   COMPUTATIONAL COST

Our method is implemented in JAX. We utilize the `vmap` to the world model i.e. ensemble members in parallel. The table below shows the wall-clock time for training world models in parallel and the time saved compared to training each one-by-one. Table 2 shows the time efficiency of using the vectorizing map with JAX. Each row shows the time for one *full epoch* of a Halfcheetah Brax training dataset of size $10^6$ samples with 23 input features and 18 output features. The model has 10 fully connected hidden layers of 256 dimensions each.

Table 2: Wall-clock time in minutes on a single NVIDIA A40

| No. models | Serial | `vmap` **(ours)** | time saved |
|---|---|---|---|
| **1** | 0.16 | 0.16 | 0.00 |
| **5** | 0.80 | 0.23 | 0.57 |
| **10** | 1.59 | 0.30 | 1.29 |
| **25** | 3.98 | 0.56 | 3.42 |
| **50** | 7.96 | 1.01 | 6.95 |

Note that if possible, our method's full implementation in JAX allows for the use of `pmap` to parallelize across GPUs which would cut linearly reduce the time on column by the number of available GPUs. This is not require for our method, a single GPU is sufficient to reproduce the entire pipeline.

## A.4 World Model Training Results

The results after training the world models and testing on held-out sequences. D4RL data obtained from (Fu et al., 2020) and the visual D4RL from Lu et al. (2023).

Table 3: $L_2$ loss in world model training results for different $\mathcal{D}$ ratios across environment

| Environment | % of $|\mathcal{D}|$ | Train Loss Mean | Train Loss Median | Test Loss Mean | Test Loss Median |
|---|---|---|---|---|---|
| Pendulum-v1 | 1 | $1.201 \cdot 10^{-7}$ | $1.19 \cdot 10^{-7}$ | $5.87 \cdot 10^{-4}$ | $5.83 \cdot 10^{-4}$ |
| | 5 | $2.20 \cdot 10^{-6}$ | $2.19 \cdot 10^{-6}$ | $5.93 \cdot 10^{-5}$ | $5.91 \cdot 10^{-5}$ |
| | 10 | $4.28 \cdot 10^{-6}$ | $4.39 \cdot 10^{-6}$ | $3.02 \cdot 10^{-5}$ | $3.01 \cdot 10^{-5}$ |
| | 20 | $6.85 \cdot 10^{-6}$ | $6.90 \cdot 10^{-6}$ | $1.87 \cdot 10^{-5}$ | $1.86 \cdot 10^{-5}$ |
| | 50 | $9.35 \cdot 10^{-6}$ | $9.34 \cdot 10^{-6}$ | $1.33 \cdot 10^{-5}$ | $1.34 \cdot 10^{-5}$ |
| | 70 | $3.99 \cdot 10^{-1}$ | $1.02 \cdot 10^{-5}$ | $4.08 \cdot 10^{-1}$ | $1.28 \cdot 10^{-5}$ |
| | 100 | $3.99 \cdot 10^{-1}$ | $1.11 \cdot 10^{-5}$ | $4.08 \cdot 10^{-1}$ | $1.23 \cdot 10^{-5}$ |
| Acrobot | 1 | $8.86 \cdot 10^{-7}$ | $9.11 \cdot 10^{-7}$ | $1.20 \cdot 10^{-2}$ | $1.20 \cdot 10^{-2}$ |
| | 5 | $7.53 \cdot 10^{-6}$ | $7.35 \cdot 10^{-6}$ | $2.55 \cdot 10^{-3}$ | $2.57 \cdot 10^{-3}$ |
| | 10 | $1.71 \cdot 10^{-5}$ | $1.69 \cdot 10^{-5}$ | $1.17 \cdot 10^{-3}$ | $1.18 \cdot 10^{-3}$ |
| | 20 | $3.37 \cdot 10^{-5}$ | $3.37 \cdot 10^{-5}$ | $5.05 \cdot 10^{-4}$ | $5.05 \cdot 10^{-4}$ |
| | 50 | $7.60 \cdot 10^{-5}$ | $7.60 \cdot 10^{-5}$ | $3.01 \cdot 10^{-4}$ | $3.02 \cdot 10^{-4}$ |
| | 70 | $9.14 \cdot 10^{-5}$ | $9.09 \cdot 10^{-5}$ | $2.67 \cdot 10^{-4}$ | $2.66 \cdot 10^{-4}$ |
| | 100 | $1.40 \cdot 10^{-4}$ | $1.39 \cdot 10^{-4}$ | $2.81 \cdot 10^{-4}$ | $2.81 \cdot 10^{-4}$ |
| Cartpole | 1 | $1.95 \cdot 10^{-8}$ | $1.86 \cdot 10^{-8}$ | $3.57 \cdot 10^{-5}$ | $3.60 \cdot 10^{-5}$ |
| | 5 | $2.97 \cdot 10^{-7}$ | $2.89 \cdot 10^{-7}$ | $4.20 \cdot 10^{-6}$ | $4.15 \cdot 10^{-6}$ |
| | 10 | $4.86 \cdot 10^{-7}$ | $4.85 \cdot 10^{-7}$ | $2.22 \cdot 10^{-6}$ | $2.23 \cdot 10^{-6}$ |
| | 20 | $6.49 \cdot 10^{-7}$ | $6.47 \cdot 10^{-7}$ | $1.52 \cdot 10^{-6}$ | $1.52 \cdot 10^{-6}$ |
| | 50 | $8.05 \cdot 10^{-7}$ | $8.03 \cdot 10^{-7}$ | $1.15 \cdot 10^{-6}$ | $1.14 \cdot 10^{-6}$ |
| | 70 | $8.61 \cdot 10^{-7}$ | $8.61 \cdot 10^{-7}$ | $1.08 \cdot 10^{-6}$ | $1.08 \cdot 10^{-6}$ |
| | 100 | $8.98 \cdot 10^{-7}$ | $8.98 \cdot 10^{-7}$ | $1.05 \cdot 10^{-6}$ | $1.04 \cdot 10^{-6}$ |
| Hopper | 1 | $1.88 \cdot 10^{-3}$ | $1.98 \cdot 10^{-3}$ | $1.04 \cdot 10^{-2}$ | $8.79 \cdot 10^{-3}$ |
| | 5 | $1.47 \cdot 10^{-3}$ | $1.01 \cdot 10^{-3}$ | $9.09 \cdot 10^{-3}$ | $8.05 \cdot 10^{-3}$ |
| | 10 | $1.21 \cdot 10^{-3}$ | $2.30 \cdot 10^{-4}$ | $8.15 \cdot 10^{-3}$ | $7.40 \cdot 10^{-3}$ |
| | 25 | $1.08 \cdot 10^{-3}$ | $3.21 \cdot 10^{-4}$ | $7.41 \cdot 10^{-3}$ | $6.24 \cdot 10^{-3}$ |
| | 50 | $9.71 \cdot 10^{-4}$ | $3.32 \cdot 10^{-4}$ | $6.82 \cdot 10^{-3}$ | $5.10 \cdot 10^{-3}$ |
| | 75 | $8.87 \cdot 10^{-4}$ | $3.16 \cdot 10^{-4}$ | $6.31 \cdot 10^{-3}$ | $4.79 \cdot 10^{-3}$ |
| | 100 | $8.20 \cdot 10^{-4}$ | $3.02 \cdot 10^{-4}$ | $5.91 \cdot 10^{-3}$ | $4.36 \cdot 10^{-3}$ |
| Halfcheetah | 1 | $4.3 \cdot 10^{-3}$ | $3.8 \cdot 10^{-3}$ | $2.3 \cdot 10^{-2}$ | $2.0 \cdot 10^{-2}$ |
| | 5 | $3.4 \cdot 10^{-3}$ | $1.9 \cdot 10^{-3}$ | $1.9 \cdot 10^{-2}$ | $1.6 \cdot 10^{-2}$ |
| | 10 | $2.8 \cdot 10^{-3}$ | $5.6 \cdot 10^{-4}$ | $1.6 \cdot 10^{-2}$ | $1.3 \cdot 10^{-2}$ |
| | 25 | $2.4 \cdot 10^{-3}$ | $5.2 \cdot 10^{-4}$ | $1.3 \cdot 10^{-2}$ | $9.2 \cdot 10^{-3}$ |
| | 50 | $2.1 \cdot 10^{-3}$ | $4.9 \cdot 10^{-4}$ | $1.2 \cdot 10^{-2}$ | $5.5 \cdot 10^{-3}$ |
| | 75 | $1.9 \cdot 10^{-3}$ | $4.7 \cdot 10^{-4}$ | $1.1 \cdot 10^{-2}$ | $4.6 \cdot 10^{-3}$ |
| | 100 | $1.7 \cdot 10^{-3}$ | $4.2 \cdot 10^{-4}$ | $9.5 \cdot 10^{-3}$ | $3.8 \cdot 10^{-3}$ |
| Hopper D4RL | 10 | $6.07 \cdot 10^{-4}$ | $6.12 \cdot 10^{-4}$ | $1.27 \cdot 10^{-3}$ | $1.27 \cdot 10^{-3}$ |
| | 25 | $6.10 \cdot 10^{-4}$ | $6.09 \cdot 10^{-4}$ | $1.08 \cdot 10^{-3}$ | $1.08 \cdot 10^{-3}$ |
| | 50 | $5.96 \cdot 10^{-4}$ | $5.97 \cdot 10^{-4}$ | $9.76 \cdot 10^{-4}$ | $9.75 \cdot 10^{-4}$ |
| | 75 | $6.47 \cdot 10^{-4}$ | $6.47 \cdot 10^{-4}$ | $9.48 \cdot 10^{-4}$ | $9.47 \cdot 10^{-4}$ |
| | 100 | $6.84 \cdot 10^{-4}$ | $6.85 \cdot 10^{-4}$ | $9.10 \cdot 10^{-4}$ | $9.09 \cdot 10^{-4}$ |
| Halfcheetah D4RL | 10 | $9.30 \cdot 10^{-4}$ | $9.29 \cdot 10^{-4}$ | $5.86 \cdot 10^{-3}$ | $5.87 \cdot 10^{-3}$ |
| | 25 | $7.32 \cdot 10^{-4}$ | $7.32 \cdot 10^{-4}$ | $3.66 \cdot 10^{-3}$ | $3.66 \cdot 10^{-3}$ |
| | 50 | $5.48 \cdot 10^{-4}$ | $5.46 \cdot 10^{-4}$ | $2.50 \cdot 10^{-3}$ | $2.50 \cdot 10^{-3}$ |
| | 75 | $4.46 \cdot 10^{-4}$ | $4.46 \cdot 10^{-4}$ | $1.99 \cdot 10^{-3}$ | $1.99 \cdot 10^{-3}$ |
| | 100 | $3.86 \cdot 10^{-4}$ | $3.85 \cdot 10^{-4}$ | $1.69 \cdot 10^{-3}$ | $1.69 \cdot 10^{-3}$ |

Table 4: $L_2$ loss for the visual model

| Environment | % of $|\mathcal{D}|$ | Train Loss Mean | Train Loss Median | Test Loss Mean | Test Loss Median |
|---|---|---|---|---|---|
| cheetah-run | 100 | $2 \cdot 10^{-3}$ | $2 \cdot 10^{-3}$ | $8.2 \cdot 10^{-3}$ | $8.1 \cdot 10^{-3}$ |

## A.5 HYPERPARAMETERS

Hyperparameters for our method.

Table 5: Hyperparameters for the world model training

| Hyperparameter | Value |
|---|---|
| Learning Rate | $1 \cdot 10^{-4}$ |
| Batch Size | 64 |
| Hidden Size | 256 |
| Epochs | 400 |

Table 6: Hyperparameters for the visual world model training

| Hyperparameter | Value |
|---|---|
| Learning Rate | $1 \cdot 10^{-4}$ |
| Batch Size | 8 |
| Epochs | 100 |
| Encoder Hidden Dims | (64, 128, 256) |
| Encoder Kernel Size | (3, 3) |
| Encoder Stride | (2, 2) |
| Decoder Initial Size | (8,8) |
| Decoder Kernel Size | (4, 4) |
| Decoder Stride | (2, 2) |
| Padding | SAME |
| Dynamics Hidden Size | 256 |
| Reward Predictor Hidden Size | 256 |
| Input Image Size | (64, 64, 3) |
| Output Image Size | (64, 64, 3) |

Table 7: Hyperparameters for Each RL Environment

| Hyperparameter | Acrobot | CartPole | Hopper | HalfCheetah | Pendulum |
|---|---|---|---|---|---|
| Learning Rate | $5 \cdot 10^{-4}$ | $2.5 \cdot 10^{-4}$ | $3 \cdot 10^{-4}$ | $1 \cdot 10^{-3}$ | $1 \cdot 10^{-3}$ |
| Number of Environments | 16 | 4 | 512 | 16 | 32 |
| Total Timesteps | $5 \cdot 10^5$ | $5 \cdot 10^5$ | $5 \cdot 10^7$ | $5 \cdot 10^7$ | $1 \cdot 10^7$ |
| PPO Update Epochs | 4 | 4 | 4 | 64 | 4 |
| Number of Minibatches | 4 | 4 | 32 | 4 | 4 |
| Gamma | 0.99 | 0.99 | 0.99 | 0.99 | 0.99 |
| GAE Lambda | 0.95 | 0.95 | 0.95 | 0.95 | 0.95 |
| Clip EPS | 0.2 | 0.2 | 0.2 | 0.2 | 0.2 |
| Entropy Coefficient | 0.01 | 0.01 | 0.0 | 0.003 | 0.01 |
| Value Function Coef | 0.5 | 0.5 | 0.5 | 0.5 | 0.5 |
| Max Grad Norm | 1 | 0.5 | 0.5 | 1 | 1.0 |
| Activation Function | tanh | tanh | tanh | tanh | tanh |
| Anneal Learning Rate | true | true | false | true | true |
| Number of Eval Envs | 1 | 1 | 1 | 1 | 1 |
| Eval Frequency | 4 | 4 | 100 | 4 | 4 |

Table 8: Hyperparameter range sweep for SAC N

| Hyperparameter | Values |
|---|---|
| polyak step size | [0.004, 0.006] |
| gamma | 0.99, 0.999 |
| lr | $5 \times 10^{-5}, 1 \times 10^{-4}, 2 \times 10^{-4}, 3 \times 10^{-4}$ |
| num of critics | 200, 300, 500 |
| batch size | 128, 256, 512 |

## A.6 FURTHER ENSEMBLE SIZE ABLATIONS

Here we present the ablations for the classic control environments.

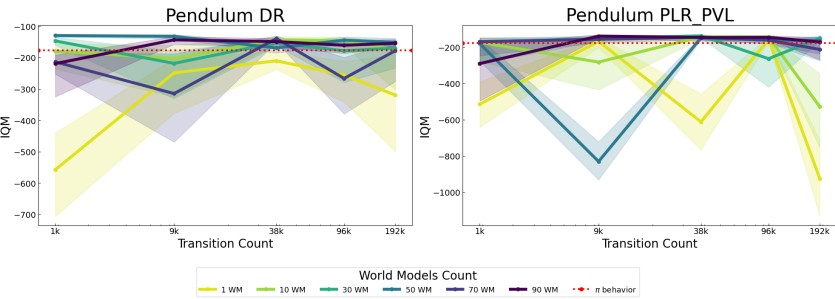

Figure 13: Ablations for Pendulum-v1

Table 9: Hyperparameter range sweep for CQL

| Hyperparameter | Values |
|---|---|
| polyak step size | [0.004, 0.006] |
| gamma | 0.99, 0.999 |
| lr | $5 \times 10^{-5}, 1 \times 10^{-4}, 3 \times 10^{-4}$ |
| num critics | 200, 300, 500 |
| batch size | 128, 256, 512 |
| seed | 1, 2, 3 |
| cql target actions gap | [0.5, 2.0] |
| cql temperature | [0.5, 2.0] |
| cql min q weight | [1.0, 10.0] |
| cql n actions | 5, 10, 15 |

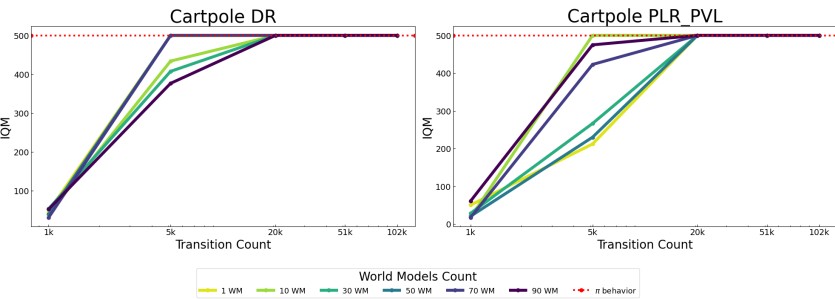

Figure 14: Ablations for symbolic Cartpole

## A.7 HIDDEN STATES VISUALIZATION

Each row illustrates the episodic progression, with Figure 15 depicting the 2-dimensional Principal Component Analysis (PCA) of the 256-dimensional hidden states. These hidden states are collected from 10 differently initialized rollouts of the same agent. The rollouts are performed across 9 different world models and the real environment, ensuring a fair and balanced classification dataset. Notably, no pattern of stability emerges with the **DR**-trained agent. However, the **PLR** and **PLR_PVL** agents exhibit stabilization midway through the episode, within a smaller range on the principal components compared to the PCA of their initial state. While this warrants further investigation, we can intuitively infer that the agent learns to act optimally across all world models, and that this optimal behavior tends to become increasingly similar—**though still distinct**—across the different world models and environments.

## A.8 HIDDEN STATES CLASSIFICATION

Table 10: Classification accuracy of 9 world models and the real environment

| % of $|\mathcal{D}|$ | DR | PLR | PLR_PVL |
|---|---|---|---|
| 1 | 0.68 | 0.11 | 0.47 |
| 5 | 0.41 | 0.65 | 0.67 |
| 10 | 0.62 | 0.68 | 0.40 |
| 20 | 0.67 | 0.67 | 0.09 |
| 50 | 0.76 | 0.66 | 0.36 |
| 70 | 0.68 | 0.58 | 0.37 |
| 100 | 0.54 | 0.85 | 0.79 |

The confusion matrix for the classification of the world model using the agent's recurrent state from all the steps of the episode.

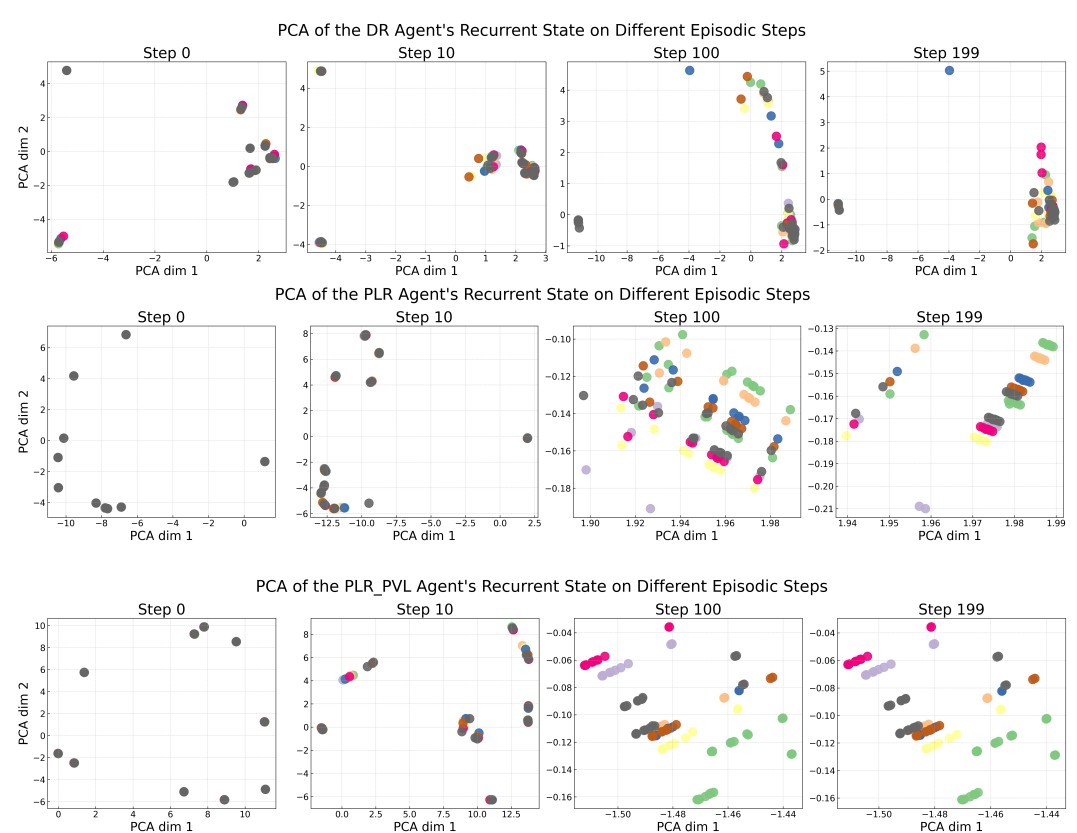

Figure 15: PCA of the hidden recurrent state for agents trained on different algorithms

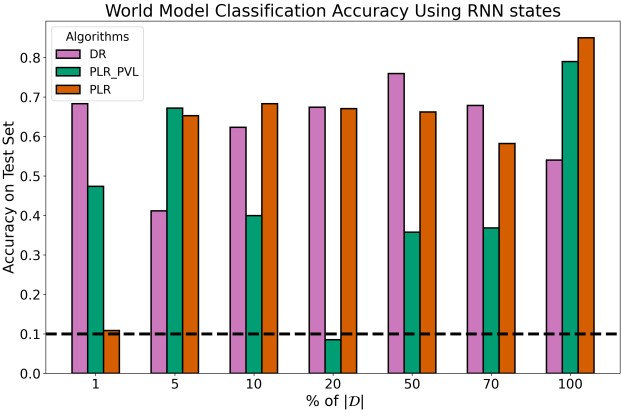

Figure 16: Classification accuracy of the hidden states from agents trained with DR, PLR, and PLR_PVL for a dataset of trajectories from 9 world models and the real environment. The dashed black line is the random prediction accuracy for the 10 classes.

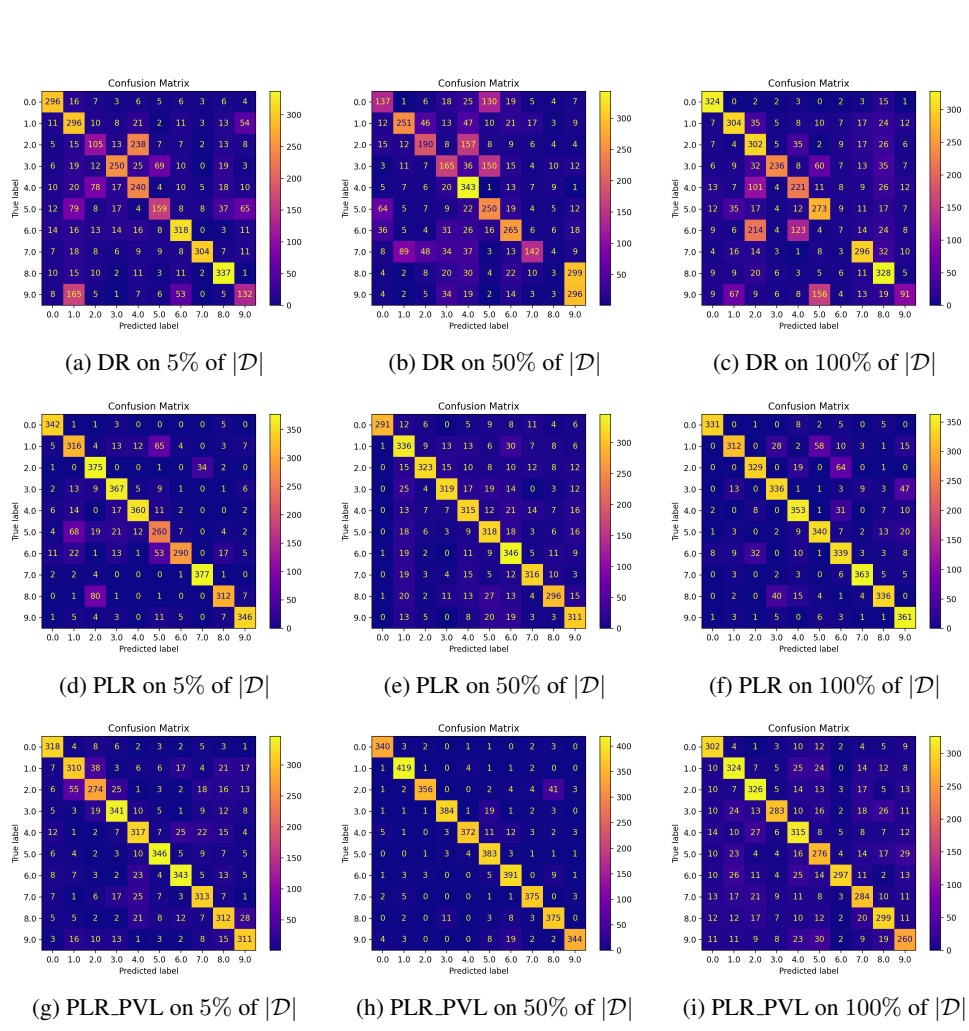

(a) DR on 5% of $|\mathcal{D}|$    (b) DR on 50% of $|\mathcal{D}|$    (c) DR on 100% of $|\mathcal{D}|$

(d) PLR on 5% of $|\mathcal{D}|$    (e) PLR on 50% of $|\mathcal{D}|$    (f) PLR on 100% of $|\mathcal{D}|$

(g) PLR_PVL on 5% of $|\mathcal{D}|$    (h) PLR_PVL on 50% of $|\mathcal{D}|$    (i) PLR_PVL on 100% of $|\mathcal{D}|$

Figure 17: Confusion Matrix for classifying 10 different levels or training environments using the RNN hidden states. Label 0 corresponds to the real **Pendulum** environment. Every row is a different training method where, DR is Domain Randomization, PLR is Prioritized Level Replay with an $L_1$ value loss score function and PLR_PVL refers to Prioritized Level Replay with an Positive Value Loss score function.

