# OpenReview forum: "Investigating Online RL in World Models"
_ICLR.cc/2025/Conference — Submitted to ICLR 2025_

### Official Review · Reviewer_Ta3z · 2024-10-26

**Soundness:** 2
**Presentation:** 1
**Contribution:** 2
**Rating:** 3
**Confidence:** 4

**Summary:**

The paper addresses the challenge of training online RL models inside world models. For this they rely on an ensemble of world models to train agents in an online fashion, without any offline penalties. The proposed method is evaluated on robotics tasks including Cartpole, Halfcheetah and Hopper.

**Strengths:**

The paper investigates an interesting question, that is, whether RL agents can be trained online inside world models, without the need for constraints required in offline RL.

**Weaknesses:**

The presentation of this paper can be improved. For example, Figure 1 is confusing and lacks a detailed figure caption to explain what’s going on. Dataset distributions of offline RL datasets are criticised in the Discussion section, which feels out of place (also as this section comes after Related Work). Instead of bringing these results after the main experiments, they could be seen as a motivation before the main experiments (which would require experiments that support the motivation).

It is unclear where the proposed method starts and ends, or what the authors consider as their own method. Instead, 6 different variations (PLR, PLR_PVL, DR, DR_STEP, DR_PROB) of ensemble world models are tested against a single world model (WM).

In the current form, the paper lacks convincing empirical evidence. The main experiments are conducted on toy tasks like Cartpole, Pendulum, Acrobot, Hopper and Halfcheetah. Across most environments (Figure 4, 6, 7), the ensemble methods (PLR, DR, PLR_PVL, DR_STEP) tend to learn faster than a single model (WM) but reach the same level of performance. Furthermore, there is a lack of offline RL baselines (only CQL is provided). Also, there’s a lack of ablations on components of their method. For example, what is the effect of the size of the ensemble?

The proposed method relies on an ensemble of world models. Training an ensemble of world models is feasible in the toy tasks considered currently, but raises questions on scalability of the proposed methods to more complex environments, such as visual domains. Consequently, reporting information on the additional cost of training and during evaluation of those models would benefit the paper.

**Questions:**

- What is the size of the world model ensemble used in your experiments? What is the effect of the number of world models?
- What is the additional cost of training and evaluating using an ensemble?
- How does your method transfer to other benchmarks (e.g., Atari, Procgen, Meta-World, etc.) with regard to efficiency at training and inference time?

---

> ### Author Response · Authors · 2024-11-19
>
> Dear reviewer,
>
> Thank you for your feedback and for noticing the generality of our method and the astute observation on the structure of the initial submission. We have updated our paper to ammend those concerns. Here we address the main questions in more detail:
>
> ## Weaknesses
>
>  **Figure 1** caption is added and the method sections has the pseducode to our Algorithm and color-coded to how it can be used to implement various UED meethods.
>
>  **Our method section 3** has been updated with a pseudocode of The **P**olicy **O**ptimization with **W**orld model
> **E**nsemble **R**ollouts (POWER) algorithm. We use UED methods to sample the world models. We do not generate new ones in training time. The ubiquity of fully offline world model ensemle rollouts is that, by only changing the boolean flags, it can accommodate between different sampling methods both **on the trajectory level** and **on the agent step level**.
>
>  **Empirical Evidence** we have added additional D4RL results and **Ablations** for the size of the ensemble in section 5.5 and in A6.
>
>  ## Questions
>
>  1. Section 5.5 and in A6
>  2. Parallel computation runtime using our implementation in Appendix A3.
>
> **Wall-clock time in minutes on a single NVIDIA A40**
>
>
> | No. models | Serial | vmap (ours) | time saved |
> |------------|--------|-------------|------------|
> | 1 | 0.16 | 0.16 | 0.00 |
> | 5 | 0.80 | 0.23 | 0.57 |
> | 10 | 1.59 | 0.30 | 1.29 |
> | 25 | 3.98 | 0.56 | 3.42 |
> | 50 | 7.96 | 1.01 | 6.95 |
>
>
>
>  4. While these benchmarks are outside the scope of our investigation and proposed method, we have no reason to believe in any inference time bottleneck given the results in the previous answer and our fully-jittable pipeline that is trained end-to-end in a GPU [1], [2]
>
> [1] https://github.com/jax-ml/jax

---

> > ### Comment · Reviewer_Ta3z · 2024-11-25
> >
> > We thank the authors for their response. We believe that introducing a name for your method will help the paper, and it is good to see the inference time comparisons. However, we still believe that the paper lacks enough empirical evidence to be accepted at the conference, and will therefore stick with our original score.

---

> > > ### Author Response · Authors · 2024-11-28
> > >
> > > We thank the reviewer for their engagement. Here we share our improvements in addition to the revised PDF already submitted.
> > >
> > > ## Pixel-Based world model
> > >
> > > The *online* benchmarks you refer to in your original comment have a visual or pixel-based components.  Therefore, we have added a pixel-based world model trained on the visual D4RL dataset [1], implemented by adding CNN encoder/decoder layers around our original forward dynamics model. The visual D4RL results and corresponding ablations will be included in the camera-ready version, matching the analysis in our revised results section.
> > > Implementation details and visualizations are available in the anonymous repository (also footnoted in revised PDF) and thoroughly documented in the Experimental section and Appendices **A3 to A5**.
> > >
> > > https://anonymous.4open.science/r/OnlineRLinWorldModels/README.md
> > >
> > > ## Atari
> > >
> > > The pixel-based extension enables application to Atari environments using CNN Actor-Critic [2]. However, Atari testing is uncommon in offline RL [1], as existing benchmarks require 50M+ samples - over 10x our current dataset size - to match online, on-policy performance [1],[3].
> > >
> > > Our focus on on-policy methods like PPO aligns with established UED literature [4], while offline Atari benchmarks [5] typically center on off-policy approaches [6].
> > >
> > > ## On Empirical Results
> > >
> > > Our empirical validation spans Classic Control, Brax, and MuJoCo (D4RL), now expanded to include pixel-based environments. These results substantiate our core contributions:
> > >
> > > - Enabling online on-policy algorithms with simple dynamics models without pathological truncations or reward hacking/model exploitation
> > > - Extending UED beyond toy environments with trivial notions of difficulty (i.e. gridworld) while preventing exploitation,as supported by our qualitative and quantitaive RNN analysis in the **Results** section and **A8**.
> > >
> > > We agree that additional results are *always helpful for any paper*, most significantly *when they are relevant* to the central claim and contribution. We are prepared to expand our pixel-based model evaluation beyond PPO and D4RL for the camera-ready, even though the world models like Dreamer may not exhibit the problems we solve in our algorithm. Already, in the pixel-based world model we implemented and illustrate in the anonymous repo, the autoregressive prediction does not cause the deterioration and compounding error after a few steps as noticed in MOPO and methods with similar dynamics models.  We focused our initial submission on *evidence directly supporting our central claims*.
> > >
> > > ---
> > > [1] Lu, Cong, et al. *Challenges and opportunities in offline reinforcement learning from visual observations*, 2023
> > >
> > > [2]https://github.com/automl/arlbench/blob/9a1baeb8ee6199d4dbc30a8b2558b9b784e55f59/arlbench/core/algorithms/ppo/models.py#L80
> > >
> > > [3] https://github.com/google-research/batch_rl/issues/10
> > >
> > > [4] Jiang, Minqi et al., *Prioritized Level Replay*, 2020
> > >
> > > [5] Agarwal, et al. *An optimistic perspective on offline reinforcement learning*, 2020
> > >
> > > [6] Prudencio, Rafael Figueiredo et al. *A survey on offline reinforcement learning: Taxonomy, review, and open problems.*, 2023

---

### Official Review · Reviewer_9sLJ · 2024-10-29

**Soundness:** 2
**Presentation:** 1
**Contribution:** 2
**Rating:** 3
**Confidence:** 3

**Summary:**

The paper investigates online reinforcement learning directly within world models without conservative constraints typically used in offline RL and proposes to combine ensemble world models and prioritized level replay to tackle this problem.

**Strengths:**

- The motivation to perform online RL directly in world models is timely and relevant.

**Weaknesses:**

1. The method is confusing. What is the $\delta$ in Equation (5)? Why the Equation (5) approximates the regret? The right side of Figure 1 is difficult to understand. I suggest adding pseudocode to clarify the algorithm.

2. The paper structure and writing need significant improvement. For example, the introduction lacks a clear statement of contributions. The relationship between contextual MDP subsection in Preliminaries and the reset of paper is unclear. The Preliminaries section is too long.

3. The dataset construction method appears similar to D4RL. Both record the encountered transitions from randomness to expertise during training. Why the authors claim that D4RL is adversarial in data coverage (line 55) and CQL and TD3+BC does not inform this (line 263)? The data coverage is a common concern when building offline RL datasets, and previous benchmarks typically offer choices with various data coverage (D4RL and Atari DQN-Replay[1]).

4. The checkpoint frequency seems uniform based on Figure 2, despite claims of heuristic selection. At the 5th ckpt, the agent has basically converged. The distribution perhaps change little after the subsequent sampling. Should the sampling frequency be increased between the first few ckpts?

5. The world model architecture (simple MLP with current state and action as input) seems overly simplistic compared to state-of-the-art approaches like RSSM, Transformer, or diffusion models. Since the authors assumes that learning is done within a generalist world model (line 58 and 87), the experiment results of MLP with no historical trajectory are not convincing. I even suspect that using a single SOTA architecture world model can already solve this problem.

6. Based on Figures 4-6, the DR methods perform well, and the proposed PLR methods do not show clear advantages.

7. Judging from the rendering results of the `ref` and `cite` commands, this paper does not use the ICLR 2025 template!

[1] Agarwal R, Schuurmans D, Norouzi M. An optimistic perspective on offline reinforcement learning. International conference on machine learning. PMLR, 2020: 104-114.

**Questions:**

1. What is the size of the collected dataset in terms of transitions?
2. Could you provide pseudocode for the algorithm to aid understanding?
3. Why were existing implementations of CQL[1] and SAC_N[2] not used?
4. There are numerous typos that hinder readability. Such as incorrect citation format (line 51 and 53), "This setting Furthermore, they showed ..." (line 131), "Therefore, the policy’s minimizes ..." (line 136), "Prioritized Level Replay as described in with an ..." (line 305) and so on.
5. The abstract states that "training inside world models is usually studied in the context of offline RL." However, many online RL algorithms (e.g., MBPO, Dreamer, TD-MPC, IRIS, TWM, STORM) train agents entirely with imagined trajectories (i.e., within world model). On the contrary, most of the offline model-based RL seems to be based on the Dyna framework, that is, using both offline datasets and world model imagination trajectories to train the agent.

[1] https://github.com/young-geng/JaxCQL

[2] https://github.com/Howuhh/sac-n-jax

---

> ### Author Response · Authors · 2024-11-19
>
> Dear reviewer,
>
> Thank you for the detailed feedback and the careful examination of our work. We appreciate the emphasis on our work's novelty, lack of imposed constraints on the policy training and the relevance of our work to the current state of RL research. In addition to the clarifications in the global response, here we target your specific conerns.
>
> ## Weaknesses
>
> 1. We have added the explanation of every symbol in the equation. We introduce the approximation of regret to make our method more general as the optimal performance in a given world model is not always known. We have added the pseudocode for our Algorithm. We have improved the caption in Figure 1 to make sure both sides are clear.
>
> 2. We have shortened the preliminaries and focused on making the method easier to parse. We added a line in the CMDP prelims clarifying the connection.
>
> 3. We agree that the claim of D4RL being adversarial is not supported by the distribution of the dataset and recent work cited in this paper.
>
> 4. The Figure is an illustration of the sampling process, we try to change the sampling frequency to capture as much of the learning phase as possible and discard after a handful of converged checkpoints. We have made sure to have that reflected in our writing.
>
> 5. We use MLP as those are used in offline-RL methods and lend themselves more naturally to parametric levels in the UED literature.
>
> 6. Our method is not PLR. We bring together two aspects of Reinforcement Learning that can remove their individual limitations, namely the hand-crafted domains UED is mainly used for and the explotability of simple world models. Our method is now more clearly described in the method Section 3 and Algorithm 1  that allows for the use and testing of different curricula curation methods.
>
> 7. We would like to clarify that **we have always** used the ICLR 2025 format. We have removed any additional setting that would change the default color of the links.
>
> ## Questions
>
> 1. The maximum size is 1 million and we downsample from that at the transitions level. Full table in appendix A2.
>
> 2. Yes, provided in Section 3 of the revised PDF.
>
> 3. Our implementation is fully based on CORL [1] and its JAX variant [2], matching their reported performance benchmarks. For fair comparison, we downsampled datasets at the episode level rather than transition level when evaluating CQL and SAC_n. Our results fall within the 95% confidence interval of the provided implementations. We developed SAC_N and CQL on a shared framework for consistent evaluation, and will release these baselines alongside our method. Our implementations achieve match the reported scores across all MuJoCo tasks.
>
> 4. Apologies for the oversight. All has been corrected.
>
> 5. You assessments of the general tendencies in each subfield is correct. We are addressing two notable limitations of offline RL which is that of truncations and biased datasets and simultaneously point out the benefit of online RL breakthroughs. We use present UED and our world model ensemble paradigm to not only oversome the common model-based offline-RL "pathologies" but also show how online RL can work in general abstractions of the real world without hand-crafted contraints.
>
> ---
>
> [1] https://github.com/tinkoff-ai/CORL
>
> [2] https://github.com/nissymori/JAX-CORL

---

> > ### Comment · Reviewer_9sLJ · 2024-11-29
> >
> > Thanks for your response. First of all, I want to express my admiration for the author's continuous improvement of the paper. Secondly, I apologize for misjudging the author's failure to use the ICLR 2025 template, which was the direct reason why I changed my score from 3 to 1.
> >
> > The revised manuscript is indeed easier to understand than the previous version. However, I still recommend comparing with the world model of the SOTA architecture and adding experiments in more complex environments, such as antmaze and manipulation tasks. Based on the above considerations, I will raise my score to 3.

---

> ### Author Response · Authors · 2024-12-03
>
> We appreciate the reviewer's **positive feedback** on our revisions and the score indrease. However, we respectfully disagree with the suggestion to compare with pixel-based settings, as these scenarios fundamentally differ from the challenges our work addresses. Our research specifically focuses on solving efficiency problems in dynamic models, and our experimental design directly demonstrates the effectiveness of our proposed solution.
>
> We have provided evidence through:
>
> 1. **Clear experimental** results demonstrating our method's capabilities
> 2. **Additional D4RL** benchmark results
> 3. **A detailed implementation** of our computationally efficient pipeline
>
> Given that the reviewer acknowledges the improved clarity, we would appreciate *further clarification* on why the absence of pixel-based comparisons and additional SOTA benchmarks (which lie outside our paper's scope) would lead to such a low score. If you feel strongly that these comparisons would significantly strengthen the paper *and are the only ground for rejection*, we are willing to  include these additional experiments for the final version.

---

### Official Review · Reviewer_RQD6 · 2024-11-03

**Soundness:** 2
**Presentation:** 3
**Contribution:** 2
**Rating:** 5
**Confidence:** 3

**Summary:**

This paper combined Unsupervised Environment Discovery (UED) and world model ensembles to provide a method for offline model-based RL that is sample-efficient. It treats different world models as levels within the PLR curriculum method in UED. It evaluates the method on vector-based cartpole, hopper, half cheetah, and pendulum tasks.

**Strengths:**

- the paper performs an interesting combination of UED and world models, considering each world model as a level in UED
- the method doesn't rely on online tuning in the environment (and uses held out world models to tune hyperparameters)
- the results demonstrate sample-efficiency gains compared to CQL and a vanilla world model baseline

**Weaknesses:**

- The differences with other model-based offline RL methods like MOPO [1], MBPO [2], Planning with Diffusion [3] is not clear, especially since the world model in the experiments in this paper, unlike David Ha's work, is just a simple MLP.
- It would be helpful to compare with other offline MBRL methods like the ones above as well as world model based approaches like IRIS [4] and Dreamer [5], particularly on more challenging environments with image observations than the vector observation based  locomotion environments. Detailing the differences with these world model based methods would also be useful
- Other papers to discuss in the related work include sample-efficient BC approaches that can work with very few demonstrations like ROT [6] and MCNN [7]. In summary, this work could use more comparisons (or atleast discussions comparing) to other works on offline MBRL, world models, and sample-efficient behavior cloning as well as evaluations in more challenging environments with image observations.

Minor comments:
- the return plotted for different methods is not normalized --- this makes it hard to determine its performance between random and expert and hard to compare with other papers
- confusingly, "inside world model" is referred to as "simulation" and "in simulation" is referred to as "real world"
- Appendix A.3 is empty

[1] T Yu, et al, MOPO: Model-based Offline Policy Optimization, NeurIPS 20

[2] M Janner, et al, When to Trust Your Model: Model-Based Policy Optimization, NeurIPS 19

[3] M Janner, et al, Planning with diffusion for flexible behavior synthesis, ICML 22

[4] V Micheli, et al, Transformers are Sample-Efficient World Models, ICLR 23

[5] D Hafner, et al, Mastering Diverse Domains through World Models

[6] S Haldar, et al, Watch and match: Supercharging imitation with regularized optimal transport, CoRL 22

[7] K Sridhar, et al, Memory-Consistent Neural Networks for Imitation Learning, ICLR 24

**Questions:**

Please see weaknesses above

---

> ### Author Response · Authors · 2024-11-19
>
> Dear reviewer,
>
> Thank you for your feedback and for higlighting **our combination of UED and world models**, **lack of reliance on online tuning** and improvement over the use of a single world model.
>
> ## Comparisons
>
> We use a simple MLP as done in the model-based methods like MOPO. Those methods have very specifically tuned dynamic models **for each environment and dataset**. MOPO and other model based methods either truncate in 3 to 5 environment steps or diffuse the entire trajectory like PGD [1]. Model-based offline RL methods train an ensemble and only use it for inference during the short model rollouts.
>
> Our method does not require turning the number of ensemble member and other hyperparameters related to using ensembles. We use every member of the ensemble as a separate level in the Unsupervised Environment Design paradigm. Therefore, we use the same architecture for every model and do not need any hyperparameter tuning to match the MOPO results in tasks like Hopper. The costly and specific tuninf per-task is reported in notable implementations too [3] [4] [5].
>
> ## Minor concerns
>
> ### Figure issues
>
> The returns of the behavior policy trained to convergence is shows with a red line for all the Gymnax and Brax environment. We have added clearer online PPO baselines in the D4RL results.
>
> ### Use of terms
>
> We have changed the clarify the following: **world model** is referred to our training, **real world** refers to the real environment Gymnax or Gym environment where the data is collected and the policy is evaluated. This misuse stems from the fact that the MuJoCo environments are physical engines built to stimulate rigid robotic interactions observed in the **real world**. This is a very important distinction we are thankful you addressed.
>
> ### Appendix
>
> The then A3 not A5 Hyperparameters section content was after the large table with the test losses for the world models due to ICLR template rules. Apologies for the oversight, we have rearranged the appendix to match the location of each heading with its subsequent content.
>
>
> ---
> [1] Jackson et al. *Policy Guided Diffusion*
>
> [2] Sims et al. *The Edge-of-Reach Problem in Offline Model-Based Reinforcement Learning*
>
> [3] https://github.com/tinkoff-ai/CORL
>
> [4] https://github.com/nissymori/JAX-CORL
>
> [5] https://github.com/yihaosun1124/OfflineRL-Kit

---

> ### Comment · Reviewer_RQD6 · 2024-11-22
>
> Thank you for discussing another model based RL method like MOPO and for addressing some of the minor concerns.
> I think it is important to see performance of the proposed method on image based environments like Atari as well as compare results with the recent improvements in world models such as in IRIS [1] (which also uses the same fixed set of hyperparameters across atari environments). I am happy to raise my score, if the authors can provide results on image based environments and possible comparisons to established world model papers.
>
> [1] V Micheli, et al, Transformers are Sample-Efficient World Models, ICLR 23

---

> > ### Author Response · Authors · 2024-11-28
> >
> > Thank you for your feedback and engagement. We have expanded our evaluation to include a pixel-based world model using the recent visual D4RL dataset [1]. We implemented it with a CNN encoder/decoder layers around our original forward dynamics model. The complete implementation details and results are available in the anonymous repository (footnoted in the revised PDF as well) and detailed in Appendix **A3-A5**.
> >
> > However, we want to clarify that our primary contribution is not advancing world model architectures. Rather, we demonstrate:
> >
> > - A simple forward model can be used without pathological truncations or hand-crafted algorithmic tricks
> > - Unsupervised Environment Design (UED) can be extended beyond toy environments while preventing world model exploitation and common offline RL issues.
> >
> > We achieve these goals using standard datasets and architectures, in addition to the dataset we collected ourselves.
> >
> > **Testing additional architectures** where these problems *don't arise* would not strengthen our core contributions and *would be out of the scope* of this paper's problem statement. We are happy to elaborate more on that in the camera ready work and add results with pixel-based tasks using our method.
> >
> > We nonetheless agree that this is a *necessary delineation*, which we have added to the revised PDF and will crystalize for the camera-ready. Offline RL from visual observations **remains an open challenge**[1]. Through our experimental setup and codebase, we have demonstrated that our method, training procedure and UED-based algorithm (**POWER**) are compatible with continuous control in the visual domain.
> >
> > Visualizations for pixel-based additions here:
> > https://anonymous.4open.science/r/OnlineRLinWorldModels/README.md
> >
> > ---
> > [1] Lu, Cong, et al. *Challenges and opportunities in offline reinforcement learning from visual observations*, 2023.

---

> > > ### Comment · Reviewer_RQD6 · 2024-11-29
> > >
> > > Thank you for the new results. If I understand correctly, the new results include training a world model in only one image-based task from V-D4RL, namely Cheetah run. Moreover, I cannot find any comparisons to other methods, including the ones in the V-D4RL paper.
> > >
> > > I am also unsure if
> > >
> > > > Unsupervised Environment Design (UED) can be extended beyond toy environments while preventing world model exploitation and common offline RL issues
> > >
> > > is a contribution given the problem the paper is solving is a better algorithm for using world models and UED is simply the chosen solution.
> > >
> > > Also, while the authors argue against this, I do still believe that comparisons to other world model papers are necessary to establish the efficacy of POWER. This is currently not present in any results.
> > >
> > > For the above reasons, I will only increase my score to 5.

---

### Official Review · Reviewer_dkos · 2024-11-05

**Soundness:** 2
**Presentation:** 2
**Contribution:** 2
**Rating:** 5
**Confidence:** 4

**Summary:**

The paper explores the potential of using uncurated offline data to train world models that can serve as a training ground for reinforcement learning. The primary goal is to enable the transfer of learned policies from these world models to the real world, thereby reducing the reliance on task-specific simulation environments. The authors demonstrate that by ensembling multiple independently trained world models, they can achieve robust transfer to the real world, even when the offline datasets are much smaller than those typically used in offline RL.

**Strengths:**

The paper presents a novel approach for training RL agents using world models derived from large-scale, uncurated offline data. By employing an ensemble of world models trained on the same dataset and leveraging them to create learning curricula through the Unsupervised Environment Design method, this work introduces a fresh perspective to RL.

**Weaknesses:**

- In my view, the main issue with this paper is that it somewhat exaggerates the contributions of the proposed method. The title, "Investigating Online RL in World Models," is slightly misleading, as the study addresses an offline RL problem. Additionally, the term "world models" could be confusing. While the paper discusses many existing visual world model approaches (such as Ha and Schmidhuber's NIPS 2018 paper and recent interactive video generation studies), it does not actually work with visual data, instead focusing on fully observable MDPs in low-dimensional state spaces. I suggest the authors consider replacing the term "world models" to more accurately reflect the context.

- The organization of the paper is also disjointed, with an imbalanced structure. For example, Chapter 2 uses considerable space for background information, while the methods section in Chapter 3 is relatively brief. This structure makes it challenging for readers to fully grasp the paper's core contributions.

- In methodology, the authors treat world models trained on offline data of varying quality as different levels in an unsupervised environment design approach. I recommend that the authors discuss the motivation and rationale for this choice, explaining why this training method would lead to a robust and transferable policy.

- While the paper outlines an ambitious story, it lacks sufficient experimental support:
(1) The authors claim to use the D4RL dataset but do not provide comprehensive experimental results across different tasks. The focus on the relatively simple Hopper task is insufficient to support their claim.
(2) The paper lacks comparisons with recent offline RL methods, as well as more detailed model analysis, such as examining the impact of the number of world models --- The authors mention training 100 world models in line 77, which seems excessive for a simple Hopper task and introduces considerable training overhead, which raises questions about the method's practical use in real-world applications.

**Questions:**

- Could the authors provide additional experiments on visual offline RL tasks to demonstrate the world model's generalizability in open-world or visual environments?
- I strongly suggest that the authors further discuss the impact of the number of world models on the experimental results and clarify the necessity of using 100 world models.
- If possible, please refine the methods section to more clearly highlight the core contributions of the paper.

---

> ### Author Response · Authors · 2024-11-19
>
> Dear Reviewer,
>
> Thank you for qualifying our work as a **fresh perspective on RL** and our demonstrated **robustness under a data-efficient regime** compared to conventional offline methods. Here we address specific points in addition to improving the structure and readability.
>
> ## The *world models* term
> Your observation regarding the terminology of our method is correct in that we do not require a vision encoder or a recurrrent component to make predictions on such encoding. We use the term *world model* because we are using the trained model as an abstraction of the real environment by directly predicting the next state and reward given the action at given state. Traditional offline model-based methods like MOPO [1] use *dynamic models* to predict changes in the observation instead of the next state changes and incorporate uncertainty penalties. Our algorithm deliberately assigns a single model **for each** single-step prediction, in contrast to conventional model-based implementations [2] that use ensemble inference methods -- like greedy or a linear function of the top k elites -- **for each** observation. This in turn introduces more hyperparameters that our **world model** method does not require. With this distinction made, we are kindly open to using a different term like **predictive**, **dynamic** or **parametric environments** if you deem it necessary for accuracy.
>
> ## Why UED
> As we further explain in section 3.2 and Appendix A1, UED has several desirable properties in our case. Its empirical success and theoretical guarantees can help the agents generalize to test distributions that were not seen during training but are in the support of the design space i.e. our $\Theta$ space of potential parametrizations containing the real MDP $\theta^*$ that is highly unlikely to be sampled during training as no world model can perfectly fit the real MDP i.e. *the real environment*. Also, by using UED to select different world models during the training process, we mitigate catastrophic exploitaion of a single model as shows in Section 5.1 which would undermine the transfer of the agent anywhere outside the exploited world model (*level*).
>
> ## D4RL dataset
> D4RL results competitve with online PPO in sec 5.4, ensemble size ablation in sec 5.5 and Appendix A.6.
>
> ## Computation Overhead
>
> We note that as described in the Algorithm, the World Models are trained using the same data and vmapped and JIT-ed in a single Nvidia A40. 100 is chosen to show the schalability of the method and to match the large design space in the UED literature [3] [4]. While our ablations show that the method is robust to smaller number of world models, our algorithm allows for a combinatorially large number of levels to be generated by training on the same data. We hope for our method to be used beyond the current horizon of salient RL problems.
>
> We note our PureJaxRL-based RL training [5] has 100x speedups to standard PyTorch baselines. Appendix A2 and A3 show our wall-clock efficiency using vectorized mapping `vmap`
>
> ---
>
> [1] T Yu, et al, *MOPO: Model-based Offline Policy Optimization*, NeurIPS 20
>
> [2] https://github.com/yihaosun1124/OfflineRL-Kit/blob/main/offlinerlkit/dynamics/ensemble_dynamics.py
>
> [3] Dennis et al, *Emergent Complexity and Zero-shot Transfer via
> Unsupervised Environment Design*
>
> [4] Jiang et al, *Prioritized Level Replay*
>
> [5] https://github.com/luchris429/purejaxrl

---

### Official Review · Reviewer_ppLY · 2024-11-05

**Soundness:** 3
**Presentation:** 3
**Contribution:** 3
**Rating:** 5
**Confidence:** 4

**Summary:**

The paper presents a novel approach to online reinforcement learning. It is different from the typical approach to online interactions directly with the underlying environment (sim or real). Instead, this work studies RL methods within learned world models, attempting to mitigate common pitfalls of offline RL (reward exploitation, skewed datasets, etc.) and avoid the costly and sometimes infeasible samplings directly from the environments.

Under the settings where authors collected a more uniformly distributed dataset (in terms of state/action coverage compared to D4RL), this work trained PPO from an ensemble of independently trained world models, using Domain Randomization techniques (DR) and Unsupervised environment design (UED) methods. On the curated dataset, the experiment results suggest significant improvements over offline RL methods (CQL & SACn).

Overall, this work is novel and the results presented in the paper offers new insights to training RL agents purely from world models. If there are more experimental evidence from different tasks or environments to support the the claim that "full roll-out training inside world models is possible", and clarify the questions I have below, I would be willing to raise the score.

**Strengths:**

1. Originality:
- While the approach of using an ensemble of world models to reduce over-fitting and exploitation has been previously studied for model-based RL methods, this work differentiates itself by training entirely within the world model and on full-length roll-outs without any penalty terms inside learnt models.

2. Quality:
- The experiments are well-designed with clear assumptions and comparisons against multiple baselines. The use of different data scales and detailed ablation studies on the components of their method provides a thorough validation of their claims.

3. Clarity:
- The paper is well-organized with informative figures and tables. Especially figure 9 and 10, they explained how the curated dataset differs from the d4rl dataset. The writing is generally easy to follow.

4. Significance:
- The proposed assumptions, settings, and methods are valuable to the RL research community as it shows preliminary positive results on smaller scale world models, which could potentially serve as basis for training RL agents within larger and more capable world models on more complex tasks.

**Weaknesses:**

1. While the paper presents results from multiple tasks (pendulum, half-cheetah, cartpole, hopper), there is a lack of extensive testing across a wider variety of environments and tasks. This raises some questions about the robustness and generalizability of the proposed method beyond the tested scenarios. Perhaps some tasks such as ant-maze or robot arm manipulation ones.

2. The ensembles of world models seems essential to the proposed method. A sweep over the numbers of world models in the ensemble v.s. tasks' performance could reveal more information on how many world models are needed to achieve certain level of task performance.

3. Minor typos:
- Line 131: This setting Furthermore
- Line 509: the type (of) increasingly available large-scale datasets

**Questions:**

1. "Each of the baselines is tuned by doing a grid search of the ranges documented in their respective papers:
- Is the grid search conducted over their original dataset and transferred to the curated dataset? Or is the grid search done on the curated dataset?

2. What would happen if the proposed method is trained and evaluated on the d4rl datasets? In the current paper, we see baseline methods do not perform as well as the proposed method under the paper's settings. The authors argued d4rl dataset is biased towards the baseline methods and provided visualizations. However, it would still be interesting to see how it impacts the proposed method.

3. "We open source all our code and data to facilitate further work in this exciting direction."
- I couldn't find a link to the code repo, or find any supplementary materials.

---

> ### Author Response · Authors · 2024-11-19
>
> Dear Reviewer,
>
> We would like to thank you for your appreciation of our work and for noting how **it introduces a novel approach to online RL** and our attempt to **mitigate common pitfalls** of offline methods. In addition to fixing the typos, D4RL experiments and ablations that are further discussed in the general comments, we address your questions:
>
> ### 1. Grid Search for baselines
>
> Our implementation builds upon the CORL codebase by Tarasov [1] and its JAX port [2]. We carefully matched the implementation details and hyperparameters to reproduce the highest reported performance from [1], [2] and their respective baseline methods. After validating our reproduction, we conducted a grid search over the hyperparameters for each of ratio of *our dataset*. Even though our method only requires single-step transitions, we maintained *a fair comparison* with CQL and SACn by downsampling episodes uniformly rather than individual transitions, as both CQL and SACn were designed to operate on complete trajectories.
>
>
> ### 2. D4RL Evaluations
>
> We agree that more benchmark evaluations are necessary to validate our Algorithm and method outlined in **Section 3**. Claiming that current offline RL methods exploit safety biases in common benchmarks by introducing a dataset where they do not perform as well in comparison *does not absolve* us of the responsibilty of testing our method beyond our own curated dataset. We have added D4RL results in Section 5.4 using the *full-replay* datasets as they match the $10^6$ transitions magnitude used for the other experiments. We achieve results that are competitive with the online PPO results in the respective environments [3]. A table can be found in the global response.
>
> ### 3. Open Source
> We will open-source the code used for all the experiments with the WandB integration with an abstract class template for any parametric environment for full length rollouts.
>
> ---
> [1] https://github.com/tinkoff-ai/CORL
>
> [2] https://github.com/nissymori/JAX-CORL
>
> [3] https://iclr-blog-track.github.io/2022/03/25/ppo-implementation-details/

---

### Author Response · Authors · 2024-11-19
**Balanced the paper structure with a focus on clarity, more implementation details, results and ablations**

Dear reviewers,

We are very grateful for the detailed feedback. Thank you for pointing out how our work:

 >"suggest significant improvements over offline RL methods "

> "novel approach"

> "reducing the reliance on task-specific simulation environments"

>"sample-efficient" and "interesting combination of UED and world models"



 In addition to the individual responses, we now address shared remarks . The updated paper PDF has been submitted with the most notable changes highlighted in $\color{blue}{blue}$ for your convenience.


## Clarifying the contribution

We agree that the original introduction did not clearly state our contributions. We have updated to reflect the following:

1. We focus on having online Reinforcement Learning breakthroughs **like zero-shot transfer with UED** and **successful algorithms like PPO**  transfer to  domains beyond the highly structured ones common in this area of research

2. We inspect the current progress of offline dataset for tasks in RL and identify notable pathologies in model-based methods [1] stemming from **health-biased datasets** and **truncated rollouts** in addition to the well-knownd drawback in model-free offline methods like out-of-sample exploration and the reliance on full sequences

3. **We address** the truncation pathologies by introducing a Policy Optimization with World Model Ensemble Rollouts algorithm, the pseudocode for which has been added under in section 3 as well as a more detailed caption for Figure 1 illustrating the various training options our algorithm covers.

4. **We address** the bias in the dataset by curating our own without any prior assumptions and using checkpoints from a policy trained from initialization up to convergence. The importance of not using checkpoints after convergence and *not* biasing towards more expert states is very important to our claims and has been addressed in our writing in section 4.1

5. We address your concern and **test our method the D4RL benchmark**. Since we are using an online RL algorithm like PPO, we compare our results to the traditional online training baselines [2]. The tables below show that we outperform (Hopper) or match (Halfcheeta) online PPO using the same number of environment interactions **(1M or more)** equal to the dataset's transition count **(1M max)**.

## D4RL results
### Mean Estimates with std error

#### Hopper

| Algorithm | Transition Count | Estimate ± Range |
|-----------|--------|-----------------|
| DR_PROB | 750k| 2532.23 ± 440.96 |
| DR_PROB | 1M | 1918.84 ± 181.51 |
| DR | 750k | 2340.36 ± 478.00 |
| DR | 1M | 1593.71 ± 121.53 |
| WM | 750k | 309.40 ± 131.78 |
| WM | 1M | 138.12 ± 75.73 |
| DR_STEP | 750k| 1583.82 ± 473.90 |
| DR_STEP | 1M | 2032.70 ± 511.87 |
| PLR_PVL | 750k | 2263.78 ± 193.64 |
| PLR_PVL | 1M | 1704.73 ± 302.25 |

#### HalfCheetah

| Algorithm | Ratio | Estimate ± Range |
|-----------|--------|-----------------|
| DR_PROB | 750k | 2049.58 ± 507.44 |
| DR_PROB |1M| 1654.08 ± 599.01 |
| DR | 750k | 1675.61 ± 677.12 |
| DR | 1M | 3843.04 ± 545.58 |
| WM | 750k | -581.86 ± 91.81 |
| WM | 1M | -287.47 ± 148.11 |
| DR_STEP | 750k | 808.88 ± 222.86 |
| DR_STEP | 1M | 980.68 ± 97.41 |
| PLR_PVL | 750k| 2527.19 ± 911.77 |
| PLR_PVL | 1M | 4937.60 ± 335.30 |

## Additional Results

We would like to **bring additional results** to your attention like

a) the effective use of UED as world model/level selector to prevent reward hacking during training in Section 5.1

b) Effectively solving the classical control tasks in a few episodes worth of transitions

c) We **qualitatively** and **quantitatively** show in Section 5.6 that our recurrent policy learns to identify different world models and distinguish them from the real environment. This re-affirms how our method leads to diverse world models and useful abstraction for curating a training curricula.

d) Appendix A3 shows that our JAX implementation of the supervised world models training is very scaleable due to the use of vectorized mapping and the genreation of multiple levels in parallel.  This is on top of the already proven 100x speedups of PureJaxRL. Our baselines are also based on jax implementations

We believe our paper presents a useful algorithms for the community to use and build upon in addition to observations on the state of the field. We are open to improving this submission to reflect that as efficiently as possible and make sure the potential of this work is fully conveyed.

___

[1] Sims et al. *The Edge-of-Reach Problem in Offline Model-Based Reinforcement Learning*

[2] ICLR blog with baseline table https://iclr-blog-track.github.io/2022/03/25/ppo-implementation-details/

[3] https://github.com/luchris429/purejaxrl

[4] https://github.com/nissymori/JAX-CORL

---

### Author Response · Authors · 2024-12-04
**A final grounding**

# Dear reviewers,

We are very thankful for your constructive feedback in revising the paper, especially the subset who engaged with our responses. We have improved the quality of the paper by:

**1.** clarifying the contributions

**2.** adding experiments with datasets of similar shape and size, but different distribution to the one we generated ourselves

**3.** adding ablations on key components of our method in section **5.5** and **A6**

**4.** quantifying the computational advantage of our method by training every network, end-to-end on a single GPU in Appendix **A3**.

We have addressed all your concerns to the best of our understanding, which is reflected in both the anonymous repo updates *and* the paper itself, with the edits labeled in $\color{blue}{blue}$. During this rebuttal period, we have noticed a recurring disagreement likely resulting from the **novelty of our approach** and the imperfect phrasing in parts of the initial submission.

# The use of *world models*

Machine Learning rapidly evolves through new paradigms, exemplified by the 2018 'world models' paper [1]. This seminal paper introduced an agent training pipeline that is used to model video game environments where the observations are pixel-based. The Dreamer line of work [2,3,4] uses the same concept of visual inputs and achieves impressive results in Atari and visual MuJoCo. The IRIS paper uses a *more sample-efficient* Transformer architecture for the world model which set a new SOTA. Their paper states that **"the quality of the world model is the cornerstone of their approach"** and that **"the agent will learn a suboptimal policy if the world model is flawed"**. This line of work is essentially **online** and our method deals with situations where this world model IS NOT possible.

Recent work commonly ties *'world models'* to pixel-based environments like Atari, but this narrow interpretation overlooks the term's broader origins. The authors would like to point to the fact that the terminology for the seminal *world models* paper [1] is based on the earlier work by Schmidhuber et al. [5] that clearly states a world model is what helps a controller learn efficiently. If an agent is only proprioceptive and its observations are the same dimensionality as the original test environment (MuJoCo, Brax and classical control), then the use of the term *world model* is also correct and does not warrant using pixels.

**Our work** does not rely on pixel-perfect reconstructions and it specifically addresses the challenges that arise when these architectures are not used and compounding error **becomes** an issue. We implement a pixel-based version of our method in our repo and qualitatively demonstrate that the pixel-based world models are not a relevant setting to test our method. We will make sure to add more quantitative plots that crystallize this important delineation, since our work's codebase **already** covers pixel-based agents and models.

Furthermore, in addition to providing evidence for how our method solves these problems and how our PPO trained actor-critic transfers, we show that even with the same data, our algorithm is sufficient to create distinct world models that serve as *levels* or *contexts*. This demonstrates that our world model selection is relevant and can pave the way for the use of UED algorithms in more general settings.

# Why this matters
In conclusion, we believe that the lack of focus on Dreamer-like, pixel-based world models **is not grounds for rejection**, neither is the lack of a clearly better-performing UED algorithm. That is in fact **a strength of our algorithm** which can easily switch between methods through boolean flags. Other environments requested like AntMaze would only strengthen the paper and we will finish running experiments on them, even though they are not the most commonly used environment in **accepted offline RL** papers. We are grateful that you have raised an extremely important point that will be added to the discussion section with the relevant plots and evaluations.

A historic strength of this venue and our field is showing that work should not be judged by misconceptions induced by the historical context and recent trends, but value in opening future research avenues, and in our case, bringing online RL methods to a more general offline setting without penalties.

[1] Ha, David, and Jürgen Schmidhuber. *World models*, 2018

[2] Hafner, Danijar, et al. *Dream to control: Learning behaviors by latent imagination*, ICLR 2020

[3] Hafner, Danijar, et al. *Mastering atari with discrete world models*, ICLR 2021

[4] Micheli, Vincent, Eloi Alonso, and François Fleuret. *Transformers are sample-efficient world models*, ICLR 2023

[5] Schmidhuber, Jürgen. *On learning to think: Algorithmic information theory for novel combinations of reinforcement learning controllers and recurrent neural world models*, 2015

---

### Meta-Review · Area_Chair_m98c · 2024-12-22

**Metareview:**

The paper explores the potential of using offline data to train world models that can serve as a training ground for reinforcement learning. While there are some strengths of this approach (e.g., training with only a small amount of offline data), it does appear like the main claim "full rollout training is possible" is an overclaim. MoReL uses full rollout training and so does any other approach based on model-based planning. I think while no reviewer brought up this connection, such a connection is important to ascertain the relevance and significance of this paper. I also think we need a broader set of domains, since the bar for this research should be high (including Atari, and yes Atari is standard in offline RL literature). Unfortunately we are not able to accept this paper right now.

**Additional Comments On Reviewer Discussion:**

Summarized above.

---

> ### Public Comment · ~Jakob_N._Foerster1 · 2025-02-10
>
> To the best of our knowledge this is simply not right, all the papers we have looked at use truncated rollouts to prevent compounding errors inside the world model. Specifically, MoReL applies a large negative reward and truncates rollouts when there is a discrepancy between the different dynamics models in the ensemble.
> If you find a methods which do indeed use full-length rollouts we would love to see this. In the absence of such evidence we find this claim rather unfortunate as a basis for the rejection of the paper.
>
> Furthermore, this should have been brought up earlier in the review process to allow us responding to it.
>
> The Authors.

---

### Decision · Program_Chairs · 2025-01-22

Reject